

# A global lake and reservoir volume analysis using a surface water dataset and satellite altimetry

Tim Busker[1,2], Ad de Roo[1], Emiliano Gelati[1], Christian Schwatke[3], Marko Adamovic[1], Berny Bisselink[1], Jean-Francois Pekel[1], Andrew Cottam[1]

[1] European Commission Joint Research Centre, Ispra (VA), I – 21027, Italy
[2] Department of Physical Geography, Utrecht University, Utrecht, 3584 CS, the Netherlands
[3] Deutsches Geodätisches Forschungsinstitut der Technischen Universität München (DGFI-TUM), Munich, 80333, Germany

*Correspondence to*: Tim Busker (tbusker@live.nl) or Ad de Roo (Ad.DE-ROO@ec.europa.eu)

**Abstract.** Lakes and reservoirs are crucial elements of the hydrological and biochemical cycle and are a valuable resource for hydropower, domestic and industrial water use and irrigation. Although their monitoring is crucial in times of increased pressure on water resources by both climate change and human interventions, publically available datasets of lakes and reservoir levels and volumes are scarce. Within this study, a time series of variation in lake and reservoir volume between 1984 and 2015 were analysed for 135 lakes over all continents by combining the JRC Global Surface Water (GSW) dataset and the satellite altimetry database DAHITI. The GSW dataset is a highly accurate surface water dataset at 30 m resolution compromising the whole L1T Landsat 5, 7 and 8 archive, which allowed for detailed lake area calculations globally over a very long time period using Google Earth Engine. Therefore, the estimates in water volume fluctuations using the GSW dataset are expected to improve compared to current techniques as they are not constrained by complex and computationally intensive classification procedures. Lake areas and water levels were combined in a regression to derive the hypsometry relationship (dh/dA) for all lakes. Nearly all lakes showed a linear regression, and 42 % of the lakes showed a strong linear relationship with an $R^2 > 0.8$ and an average $R^2$ of 0.91. For these lakes and for lakes with a nearly constant lake area (coefficient of variation < 0.008), volume variations were calculated. Lakes with a poor linear relationship were not considered. Reasons for low $R^2$ values were found to be (1) a nearly constant lake area, (2) winter ice coverage, (3) small lake sizes and (4) a predominance of no data within the GSW dataset for those lakes. Lake volume estimates were validated for 18 lakes in the U.S., Spain, Australia and Africa using in situ volume time series, and gave an excellent Pearson correlation coefficient of on average 0.97, and a normalized RMSE of 7.42 %. These results show a high potential for measuring lake volume dynamics using a pre-classified GSW dataset, which easily allows the method to be scaled up to an extensive global volumetric dataset. This dataset will help to validate large scale hydrological models, to improve regional and global flood and drought forecasting systems and to update hydropower estimations.





## 1 Introduction

Reservoirs and lakes cover a small part of the Earth's land surface (~3.7%, Verpoorter et al, 2014), but are crucial elements in the hydrological and biochemical water cycles. Reservoirs have been constructed at a rapid pace between the 1950s and 1980s, and the construction of new reservoirs will continue over the coming century (Chao et al., 2008; Duan and Bastiaanssen, 2013). Reservoirs therefore have an increasing impact on river discharges, as they are able to alter the hydrograph by storing, retaining and releasing water. They are a valuable resource for hydropower, domestic and industrial water use, wetlands and are the primary water resource for nearly half of the irrigation-based agricultural sector by supplying approximately 460 km$^3$ of water per year (Biemans et al., 2011; Hanasaki et al., 2006). Moreover, they play a crucial role in biogeochemical activity by emitting vast amounts of $CO_2$, triggered by $CO_2$ saturation in lakes and wetlands worldwide (Balmer and Downing, 2011; Cole et al., 2007; Frey and Smith, 2005; Richey et al., 2002).

The amount of water in a reservoir results from the balance of inflow - i.e. direct precipitation, inflowing river discharge, discharge from riparian communities and industries, and subsurface inflow - and outflow - i.e. direct evaporation, withdrawals, reservoir outflow and groundwater percolation - (Duan and Bastiaanssen, 2013). A long-term imbalance can result in considerable reductions in water storage, as frequently observed around the globe in for example Lake Mead, Lake Powell, Lake Poopo and the Aral Sea (Barnett and Pierce, 2008; Micklin, 2016). Reduced water availability in the reservoirs may then result in reductions in hydropower energy production and/or irrigation water availability and lead to economic and societal damage. Many studies have already pointed out that population and economic growth, together with climate change and increasing energy and food requirements will put increasing pressure on water resources (Haddeland et al., 2014; Liu, 2016). A proper understanding of the historical dynamics of reservoirs as a source of water for irrigation, drinking water and energy production, as well as a buffer for flood protection is essential to also improve the quality of future projections on global water resources.

While for individual river basin studies information on reservoirs may be available, especially for larger scale water resource studies at national, continental and global scale, almost no historical records on reservoirs are readily available to run, calibrate and validate hydrological models (Hanasaki et al., 2006). Moreover, in situ lake level and volume measurements are sparse – especially in developing countries – and have even decreased around the globe during last years (Duan and Bastiaanssen, 2013). Even if water levels or volumes are monitored, the information is rarely freely available due to strategic political, commercial or national legislation reasons. Therefore, only a few comprehensive global lake and reservoir data sets exist (e.g. Downing et al., 2006; Lehner and Döll, 2004; Meybeck, 1995; Verpoorter et al., 2014) and if they provide a water storage estimation, these estimates are not dynamic or do not provide data over a longer time series. Therefore, remotely sensed data may be a valuable alternative to monitor water volumes in lakes and reservoirs over the last few decades.

Monitoring lakes and reservoirs using remote sensing has gained much attention over the last few years (e.g. Avisse et al., 2017; Crétaux et al., 2016; Duan and Bastiaanssen, 2013; Frappart et al., 2006b; Gao et al., 2012; Smith and Pavelsky,





2009). Most of these publications focussed on volume variations by combining altimetry water level with lake area from a multispectral sensor. Landsat or MODIS imagery is commonly used to estimate water surface areas, by classifying the satellite images capturing the water body. The classification procedure is demanding and computationally intensive if large areas or many images are classified, and misclassifications may occur because of the diversity of spectral signatures emitted

by water surfaces. Therefore, calculating lake areas is often a constraining factor in lake volume calculations. They are predominantly used for the lake hypsometry relationship (dh/dA), but they normally don't provide any temporal details and therefore cannot be used to calculate volume variations on their own (e.g. Duan and Bastiaanssen, 2013; Ran and Lu, 2012; Zhang et al., 2006). Where lakes areas have been calculated to any great extent, this has only been done for a couple of lakes or at a lower resolution (e.g. Smith and Pavelsky, 2009; Tong et al., 2016). Thus, measuring lake volume variations from

space is commonly a trade-off between the number of lakes analysed, the resolution of the lake area calculation and the number of historical lake areas that can be calculated. In this study however, by using the pre-processed recently available Joint Research Centre (JRC) Global Surface Water (GSW) dataset with a high temporal and spatial resolution and extensive validation, this trade-off is no longer an issue. Here we perform volume variation estimations globally with a 30 m resolution from 1984 onwards, using all Landsat images available after the launch of Landsat 5. Thereby this study aims to improve on

current monitoring techniques and to develop an automatic methodology that is relatively easy to implement at a large scale.

The paper is organized as follows. Section 2 presents the data used in this research, providing a description of the DAHITI altimetry database and an overview of the GSW dataset. Section 3 contains a description of the methods applied, while Sect. 4 gives a description of the results. Section 5 presents a discussion, and finally the conclusions and recommendations are presented in Sect. 6.

## 2. Data

### 2.1 Satellite altimetry

Satellite altimetry was initially designed for observing the ocean`s surface. But for more than 10 years now, satellite altimetry has proven to be a suitable tool for measuring water heights of lakes and rivers. Numerous studies have already shown the potential of estimating water level time series over inland waters using different altimeter missions such as

Topex/Poseidon (Birkett, 1995), Envisat (Frappart et al., 2006a), Saral (Schwatke et al., 2015b), Cryosat-2 (Villadsen et al., 2015) or ICESat (Zhang et al., 2011). Water levels from satellite altimetry have also been used for hydrological applications such as the estimation of river discharge (Kouraev et al., 2004; Tourian et al., 2017; Zakharova et al., 2006) and lake volumes (Duan and Bastiaanssen, 2013; Tong et al., 2016; Zhou et al., 2016).

Satellite altimetry has the potential to provide reliable water level time series of globally distributed inland water bodies

over the last 20 years. However, satellite altimetry is measured only along mission-dependent ground tracks. Therefore, larger lakes and reservoirs have a higher probability to be crossed by a satellite track than smaller ones. Topex/Poseidon and Jason-1/-2/-3 have an identical orbit configuration with a 9.9156 days repeat cycle and a track separation of about 300 km at



the Equator. ERS-1/-2, Envisat and SARAL flew on an orbit with a 35 days repeat cycle and a track separation of about 80 km at the Equator. The combination of different altimeter missions is essential to increase the temporal resolution, spatial resolution and length of the water level time series. In order to combine altimeter data from different missions, a mission-dependent range bias resulting from a multi-mission crossover analysis has to be taken into account to achieve long

homogenous water level time series (Bosch et al., 2014). The estimation of water level time series for small lakes, reservoirs or rivers is very challenging for satellite altimetry because of the footprint size. In the best case, satellite altimetry has the capability to observe rivers with a width of about 100-200 m or lakes with a diameter of a few hundred meters. Over inland waters, the diameter of the footprint varies between 2 km over the ocean and up to 16 km over land (Fu and Cazenave, 2001). Land influences within the altimeter footprint can affect the altimeter waveforms and require an additional retracking

to achieve more accurate ranges. The off-nadir effect is another problem which can occur when investigating smaller water bodies. In general, satellite altimetry is measured in the nadir direction, but if the investigated water body is not located in the center of the footprint, then the radar pulses are not reflected in the nadir direction which leads to longer corrupted ranges that must be taken into account (Boergens et al., 2016).

In this paper we use water level time series from the „Database for Hydrological Time Series over Inland

Waters" (DAHITI) as input data for the volume estimation. DAHITI is an altimetry database launched in 2013 by the „Deutsches Geodätisches Forschungsinstitut der Technischen Universität München" (DGFI-TUM). The data is accessible through a user-friendly web service (http://dahiti.dgfi.tum.de/en/) and is currently providing water levels for 680 lakes, reservoirs, rivers and wetlands. The processing strategy of DAHITI is based on a Kalman filtering approach and an extended outlier detection (Schwatke et al., 2015a) which combines different altimeter missions such as Topex/POSEIDON, Jason-1, 2

and 3, GFO, Envisat, ERS-1 and 2, Cryosat-2, and SARAL/AltiKa. DAHITI uses only high-frequency altimeter data. Depending on the measurement frequency of the altimeter, heights are measured every ~620m (10Hz), ~374m (20Hz), ~294m (18Hz) or ~173m (40Hz) along the altimeter track. To achieve more accurate ranges over inland waters, the Improved Threshold Retracker (Hwang et al., 2006) is used for the reanalysis of the altimeter measurements. The DAHITI approach provides all water level time series error information based on formal errors of the Kalman filtering.

The quality of the water level time series from satellite altimetry in DAHITI has been validated with in-situ data and varies depending on the extent of the inland water body and length of the crossing altimeter track. For large lakes with ocean-like conditions such as the Great Lakes, an RMS of about 4-5 cm can be achieved. For smaller lakes and rivers an RMS of several decimeters can be achieved (Schwatke et al., 2015a).

## 2.2 The JRC Global Surface Water (GSW) dataset

The JRC Global Surface Water (GSW) dataset (Pekel et al., 2016) maps the temporal and spatial dynamics of global surface water over a 32-year period (from 16 March 1984 to 10 October 2015) at 30 m resolution. This dataset was produced by analysing the whole L1T Landsat 5, 7 and 8 archive. At the time of the study, it represented 3066080 images (1823 terabytes of data) and covered 99.95% of the landmass. The analysis was performed thanks to a dedicated expert system



classifier. The inference engine of the classifier is a procedural sequential decision tree, which used both the multispectral and multitemporal attributes of the Landsat archive as well as ancillary data layers. It assigned –in a consistent way in both space and time- each pixel to one of three target classes, either water, land or non-valid observations (snow, ice, cloud or sensor-related issues). Classification performance, measured using over 40000 reference points revealed the high accuracy of the classifier; less than 1% of false water detections, and less than 5% of omission (Pekel et al., 2016). Thanks to its technical characteristics, the GSW dataset constitutes a very valuable long-term surface water record.

The stack of classified images constitutes the long-term water history documenting the "when and where" of the water presence. This information is recorded in the monthly water history dataset – a set of 380 global scale maps documenting the water presence for each month of the 32-year archive. This monthly information constitutes the most comprehensive and detailed data set of the GSW. Eight additional information layers, documenting different facets of the surface water dynamics are also available within the GSW dataset: (1) water occurrence, (2) occurrence change intensity, (3) seasonality, (4) recurrence, (5) transitions, (6) maximum water extent, (7) monthly recurrence and (8) yearly history. In the framework of this study, the monthly water history and maximum water extent (mwe)- a map documenting places where water has been detected at least once over the 32 years- were used.

The GSW dataset was completely developed using Google Earth Engine and all of the layers are available through the Earth Engine catalog (Gorelick et al., 2017). Moreover, Earth Engine is used in this research to calculate the monthly lake area time series. Earth Engine is a cloud-based global scale platform optimized for parallel geospatial analyses and data management in earth sciences, using Google's computational power (Gorelick et al., 2017). Earth Engine allowed the analysis of lakes at global scale in high detail, while maintaining a high resolution of 30 m.

## 3 Methodology

### 3.1 Calculating monthly lake areas

Monthly time series of lake areas have been calculated for 135 lakes over all continents (Figure 1). These contained nearly all lakes available in the DAHITI altimetry database at the time of processing. No additional criteria were set for this study, as the GSW dataset covers all lakes globally. For 380 months over the period 1984-2015 lake areas were calculated using a dedicated Google Earth Engine script. For each lake, a Region of Interest (ROI) was set by a manually drawn polygon that was approximately equal to the mwe of the lake (Figure 2). For every month, lake areas were calculated directly from the GSW monthly history dataset, by counting the number of water pixels inside the polygon and multiplying this by the pixel area. To improve the accuracy of the area calculations, the amount of non-valid observations (no data pixels) within the mwe, compared to all mwe pixels within the ROI, has been expressed as the no data fraction. This no data fraction has been used to filter accurate and less accurate area observations in the regression analysis and volume calculations (see Sect. 3.2). The white striping observed for Lake Mead in Figure 2 is an example of no data that is caused by Landsat sensor issues.



Moreover, the coefficient of variation (CV) has been calculated from the area observations to express lake area variation normalized by mean lake size.

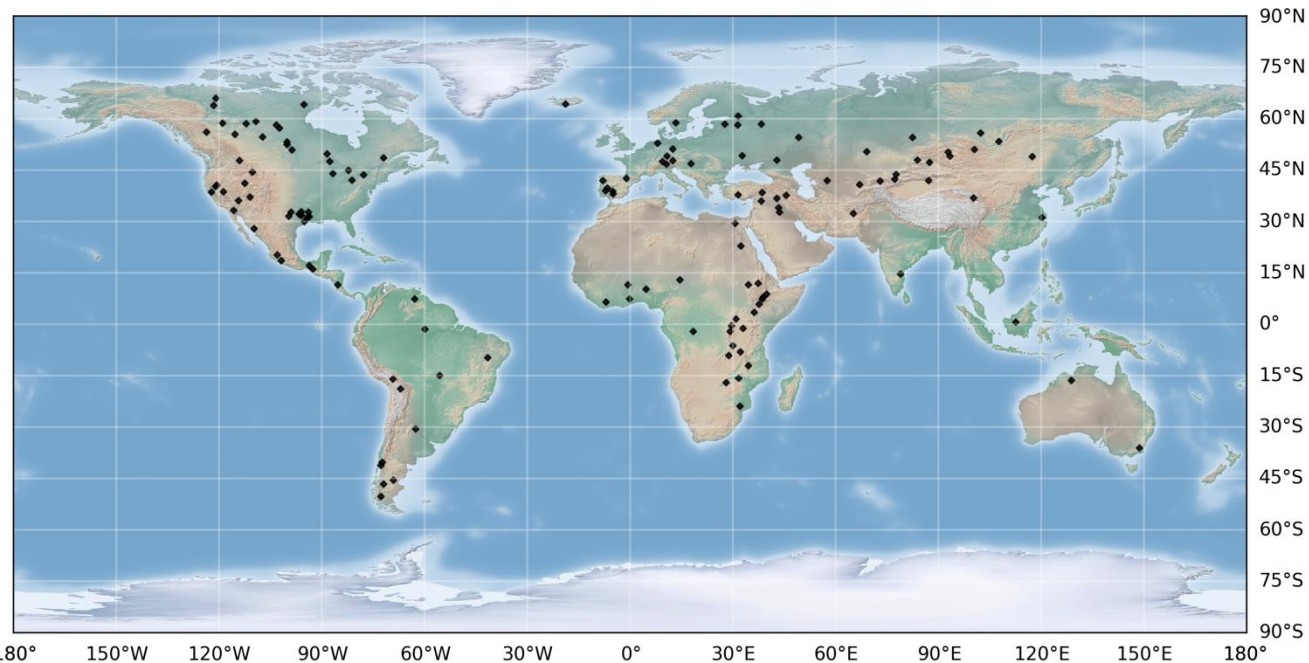

**Figure 1 Geographical distribution of the analysed lakes.**

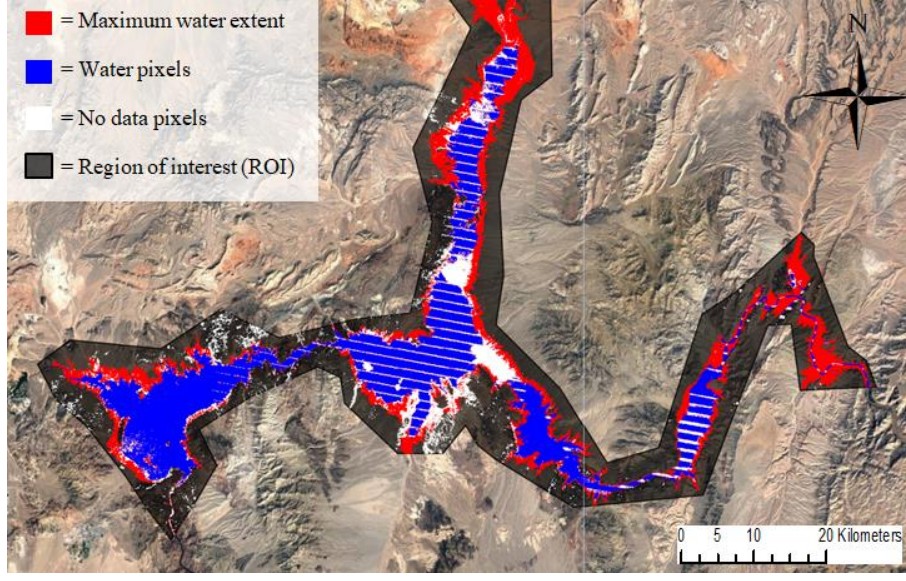

**Figure 2 An example of the area input data for Lake Mead (U.S.) for February 2015, where the maximum water extent is marked as red, water as blue and no data as white pixels.**





## 3.2 Calculating lake volumes variations

The volume of a lake or reservoir is a function of the water area ($A$) and level ($h$), derived from the hypsometry relationship (dh/dA). Monthly lake areas were only used in the regression if the no data percentage was below 1 %, because only accurate areas were desired to construct the regression line. As exact dates are not provided by the GSW dataset, the altimetry data has first been averaged per month after which monthly water levels ($h_{\text{Altimetry}}$) were coupled with monthly area values ($A_{\text{GSW}}$). These are illustrated with red dots in Figure 3 for Lake Mead (U.S.). The water level-area pairs were assumed to give linear hypsometric relationships of the water bodies (Figure 3, dashed blue line):

$$h_i = a \cdot A_i + b + \varepsilon_i \tag{1}$$

Where $A_i$ and $h_i$ are the area and water level respectively, $a$ and $b$ are the slope and intercept parameters, and $\varepsilon_i$ is the error term or residual for time step $i$. The parameters $a$ and $b$ have been derived by minimising the residual sum of squares (RSS, i.e. $\sum_i \varepsilon_i^2$), using an Ordinary Least Squares Regression (OLS) technique. Therefore, the resulting residuals have zero mean. The hypsometric relations can be integrated to obtain volume estimates of the water body:

$$V_i = \frac{(h_i - b) \cdot A_i}{2} \tag{2}$$

Where $A_i$ is the monthly calculated lake area derived from the GSW dataset, $h_i$ is the water level from altimetry and $b$ is the water level of the theoretical lake bottom from the linear regression at $A$=0, for time step $i$. By substituting the regression equation in Eq. (2), the expected value of the water volume can be calculated using $h$ or $A$ only:

$$V_i = \frac{(h_i - b)^2}{2a} = \frac{a \cdot A_i^2}{2} \tag{3}$$

Using these equations, absolute volumes are computed by using only the area or water level estimates, giving two different volumetric time series: $V_{\text{GSW}}$ and $V_{\text{altimetry}}$. Rather than a 1 % no data threshold in the regression analysis, a 5 % no data threshold has been applied to area observations used for the volume calculation. This resulted in the best trade-off between the number of observations from the GSW dataset and the accuracy of the estimates. The no data pixels within the mwe have been used to plot an upper-limit for the volume estimates, as they could theoretically be water during that specific month.

The extrapolated part of the regression line, and therefore in particular the theoretical lake bottom $b$, should be considered with caution, as the hypsometry relationship may change outside the observational range. The absolute volumes from Eq. (3) are therefore converted to volume variations from $t_0$ to $t_1$ (Figure 3, purple area). Most estimated volume variations were calculated using $h$ or $A$ values inside the observed part of the regression line (i.e. inside the range of observed h-A pairs).



However, some volumes were estimated with *A* or *h* values that are outside such range. These volume estimates use an extrapolated lake hypsometry which is not observed and therefore more uncertain. These estimates are therefore separately classified in the volumetric plots.

Not all lakes showed considerable area fluctuation, as some lakes and reservoirs are artificially bounded or have very steep banks. The signal-to-noise ratio (SNR) for lakes with a very small CV is likely to be low as errors due to no data, misclassifications and the lake border discretisation with 30 m pixels will mask the actual area variations. Therefore, the areas of lakes with a very small CV were interpreted to be constant. Lake volumes were still calculated, but only by multiplying the mean area with water level variations as observed by altimetry.

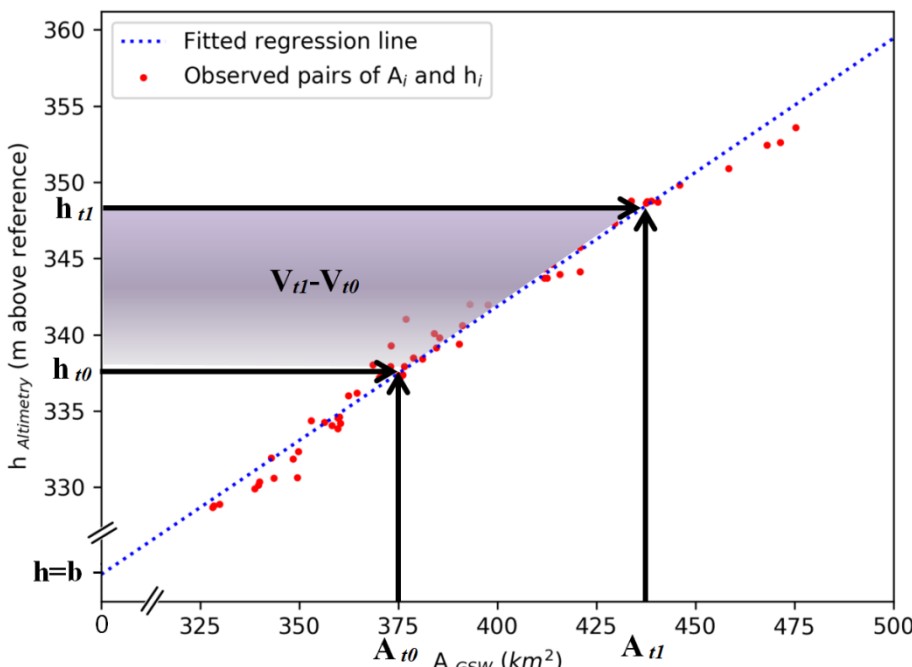

**Figure 3 Volume variation calculation for Lake Mead. The observed monthly pairs of *A* and *h* (red dots) are used to estimate a linear regression (blue dashed line) that is used to calculate volumetric changes. The volumetric change from *t₀* to *t₁* is equal to the purple area and can be visually interpreted as a 2D lake section.**

### 3.3 Validation of the volume estimates

The validation has been carried out using the Pearsons correlation coefficient $r$, the root-mean-square error (RMSE) and the normalized RMSE (NRMSE). The Pearsons correlation coefficient $r$ measures the linearity between in situ and estimated volume variations, while the RMSE accounts for the absolute error. The NRMSE has been calculated by normalizing the RMSE with the range of in situ lake volume variations:



$$NRMSE = \frac{RMSE}{Obs_{max} - Obs_{min}} \cdot 100\ \%$$
(4)

Where $Obs_{max}$ and $Obs_{min}$ are the maximum and minimum observed in situ volumes since 1984.

## 4 Results

### 4.1 Regression analysis

135 lakes and reservoirs have been analysed over all continents. The linear OLS regression analysis resulted in highly variable $R^2$ values among the lakes and reservoirs. Low $R^2$ values were observed to be caused mainly by noise rather than non-linear hypsometry, as only three lakes (Tawakoni, Urmia and Eagle) returned a clear non-linear hypsometry relationship (see discussion). The mean $R^2$ of all 135 lakes is only 0.59, but 57 lakes showed an $R^2 > 0.8$ with an average of 0.91. Lake Eucumbene (Australia), Lake Kariba (Zambia), Lake Powell, Lake Mead and Hubbard Creek Reservoir (U.S.) are examples of these lakes, with high $R^2$ values and low regression residuals. The regression results for Lake Powell, Kariba, Mead and Nasser are showed in Figure 4, with $R^2$ values of respectively 0.99, 0.96, 0.98 and 0.92.




**Figure 4 Area-level regressions for Lake Powell (a), Kariba (b), Mead (c) and Nasser (d), with $R^2$ values of respectively 0.99, 0.96, 0.98 and 0.92.**

### 4.2 Division of the lakes in groups

Based on the $R^2$ values from the regression and the area CV, the lakes have been subdivided into lakes with a constant area (Lc) and lakes with a variable area (Lv) where the latter (Lv) category has been further subdivided into lakes with a good performance (LvG; $R^2 > 0.8$) and lakes with a poor performance (LvP; $R^2 < 0.8$) (Figure 5). 41 lakes are categorized as Lc as they showed a CV < 0.008 and all these lakes returned a $R^2 < 0.6$ with an average of 0.16 (black circles, Figure 5). These lakes have a very small variation in area, resulting in highly inaccurate regressions due to a low SNR. Therefore, these lakes were assumed to have a constant area and their volume was calculated by multiplying water level observations with the average of the GSW area estimates. For CV > 0.008, lakes showed considerable monthly area variations and mostly acceptable $R^2$ values with an average of 0.76. 57 lakes where classified as LvG as they showed a $R^2 > 0.8$, with an average of





0.91 (green triangles, Figure 5). For these lakes, volumes have been calculated using accurate linear regressions. For 35 lakes with a variable area, the regression was less accurate with a $R^2 < 0.8$ and a mean of 0.52 and therefore they are classified as LvP (red triangles, Figure 5). A table for each group, showing the most important lake properties and the $R^2$ values, is given in appendix 1.

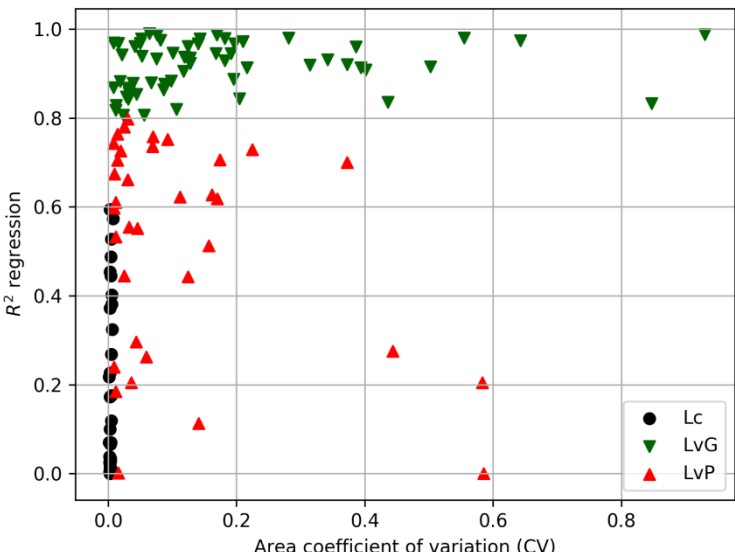

**Figure 5 Illustration of the division of the lakes into lakes with a constant area (Lc) and a variable area with a good performance (LvG) and with a poor performance (LvP), based on the $R^2$ and CV values.**

### 4.3 Volumetric results

For a total of 98 lakes (57 variable area lakes and 41 constant area lakes), the volume variation time series have been
calculated, using both water levels ($V_{\text{Altimetry}}$) and water areas ($V_{\text{GSW}}$) as inputs. Volumetric results of the variable area lakes Powell, Kariba, Mead and Nasser are outlined below and are shown in Figure 6. $V_{\text{Altimetry}}$ or $V_{\text{GSW}}$ estimates inside the observational range of the regression (*h-A* pairs) are coloured blue and red. Some volume estimates are derived from observations outside the regression range; these are extrapolated estimates and are displayed with a darker colour tint. The light red shaded areas display the uncertainty interpolated from the 5 % no data pixels in the area estimate and how this no
data propagates to the volume estimate. It represents pixels which theoretically could be water, but that were not observed due to the presence of no data within the mwe.

Figure 7 summarizes the volumetric results, by showing lake location, lake type (constant, variable good or variable poor lakes), and the average volumetric change. The volumetric change shows the magnitude of reduction (red circles) or increase (blue circles) of average water storage between the periods 1984-2000 and 2000-2015 for LvG, and from 2000-2008 to
2008-2016 for Lc. Slightly more lakes showed a positive change (59) than a negative one (37). Considerable reductions of




water storage were observed in Western U.S., due to major average volume declines in Lake Mead (10 km$^3$), Powell (6 km$^3$) and the Great Salt Lake (16 km$^3$). Average increases were found for the Great Lakes and most of the analysed lakes in Southeast Africa.

Lake Mead was formed after the construction of the Hoover Dam during the 1930s, in the former steep V-shaped slopes created by the Colorado River. It is located approximately 50 km east of Las Vegas in the Black Canyon, Arizona- Nevada (Figure 7). With a maximum depth of 158 m and a maximum capacity of 33-35 km$^3$, Lake Mead is the largest reservoir in the U.S. by capacity and the second largest (after Lake Powell) by water area (Barnett and Pierce, 2008; Holdren and Turner, 2010). The lake showed a considerable reduction in water storage between 1984 and 2015 (Figure 6c). Between two periods of maximal reported capacity (1984-1988 and 1998-2000), a small decrease in capacity of around 7 km$^3$ was observed. From its historical maximum capacity in 2000, the water level dropped 40 m between 2000 and 2010 (Cook et al., 2007; Duan and Bastiaanssen, 2013), mainly because of a combination of water abstractions by around 25 million people and multiple intensive droughts (Holdren and Turner, 2010). This reduction of water storage is also reported by the satellite estimates. According to our estimates, Lake Mead lost approximately 20 km$^3$ of water from 2000 to 2015 (Figure 6c, Figure 7). Using the USGS in situ measurement of storage volume of 31 km$^3$ in 2000, this is an almost 70 % reduction of water storage over the last 15 years.

Lake Nasser is a crucial resource for Egypt's population, functioning as a source for irrigational water and electricity and as an important flood-control mechanism. With an estimated maximum storage capacity of 162 km$^3$, Nasser is the main freshwater resource for approximately 85 % of the Egyptian population (Gao et al., 2012; Muala et al., 2014). The lake shows a strong annual cycle, with declines in water storage during the first half of the year and increases during the second half of the year (Figure 6d). The annual cycle amplitude varies from around 10 to 20 km$^3$. Besides this yearly fluctuation, Lake Nasser features an even longer inter-annual variability. From the lowest recorded volume over 1985-1993, water storage increased at least 40 km$^3$ to record-high estimates in the period 1998-2002. From 2002, the lake shows a decreasing trend towards 2006, in which it lost approximately 30 km$^3$. Two peaks were observed over 2007-2009 and in 2005, when the lake gained 30-35 km$^3$ of water and subsequently lost the same amount. The lake showed an average volume decrease over the whole observational period of only 1.8 km$^3$, which is negligible compared to the size of the lake (Figure 7).

Lake Kariba formed after the construction of the Kariba Dam on the Zambezi River on the border of Zimbabwe and Zambia (Berg et al., 1996). It has an average surface area of 5364 km$^2$ and with an estimated capacity of 185 km$^3$ it is the largest reservoir in Africa by volume (LakeNet, 2003). The reservoir shows a consistent seasonal variation, with increases of around 10-20 km$^3$ during the first five months, and decreases over the last seven months of the year (Figure 6b). Moreover, the lake gained at least 45 km$^3$ of water from 1996 to 1999. From 2000 to 2007, the volume decreased again by approximately 30 km$^3$. From 2008, the water volume increased to a maximum reported capacity in May 2010. From July 2014, the lake shows a constantly decreasing trend towards the last year of the data (September 2015). Over the whole observational period, the average storage of the lake increased by 15 km$^3$ (Figure 7).



With an area of 653 km$^2$, Lake Powell is the largest lake in the U.S. by water surface area (Barnett and Pierce, 2008; Benenati et al., 2000). Its maximum capacity of 33.3 km$^3$ is slightly less than that of Lake Mead (Benenati et al., 2000; Holdren and Turner, 2010). The lake showed two periods of maximum capacity during 1984-1988 and 1995-2000 (Figure 6a). As observed for Lake Mead, a period of intensive drought in the years after 2000 caused a considerable reduction in volume. Over the period 2000-2004, water storage declined by approximately 16 km$^3$. According to Cook et al., 2007, the volume left was only 38 % of the live capacity. The average volume decreased by 6.3 km$^3$ from 1984-2000 to 2000-2015 (Figure 7).

**Figure 6 Lake volume variations for Lake Powell (a), Kariba (b), Mead (c) and Nasser (d) using V$_{Altimetry}$ (blue) and V$_{GSW}$ (red).**




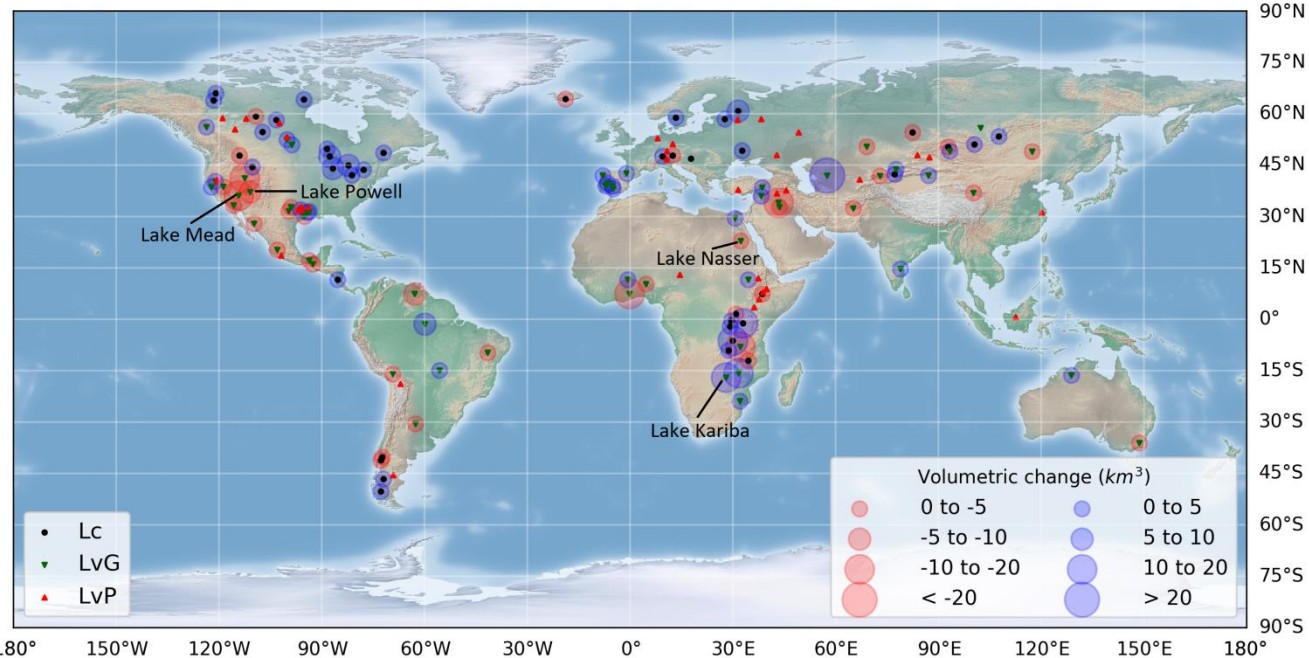

**Figure 7 Lake and reservoir types (constant (Lc), good variable (LvG) or poor variable (LvP)) with the average volume change.**

**4.4 Validation of the volume estimates**

Lake volume variations have been validated against in situ data that are based on a full bathymetric survey for 18 lakes.

Nine of these lakes are located in the USA (Richland Chambers Reservoir, Hubbard Creek Reservoir, Lake Mead, Lake Houston, Lake Powell, O. H. Ivie Lake, Toledo Bend Reservoir, Lake Walker and Lake Berryessa), 1 in Africa (Roseires Reservoir), 6 in Spain (Serena Reservoir, Puente Nuevo Reservoir, Alcantara Reservoir, Lake Almanor, Yesa Reservoir and Encoro de Salas Reservoir) and 2 in Australia (Lake Argyle and Lake Eucumbene). U.S. lakes have been validated using USGS in situ lake/reservoir volumes obtained from: https://waterdata.usgs.gov/nwis/current. Lake Powell was the only U.S.

reservoir whose data were obtained from the United States Bureau of Reclamation (USBR): https://www.usbr.gov/rsvrWater/HistoricalApp.html. Spanish validation data were gathered from the Spanish Ministry of Agriculture, Food and Environment via http://ceh-flumen64.cedex.es. In situ data for Roseires Reservoir were received from the Dams Implementation Unit of Ministry of Water Resources and Electricity, Sudan.

The validation analysis has been done for volumes both excluding and including extrapolation. The non-extrapolated

volumes showed average Pearsons $r$, RMSE and NRMSE of 0.96, 0.14 km$^3$ and 7.25 % respectively (Table 1). The high correlation indicates that the validation data show a very high linearity. The average NRMSE is relatively low taking into account all sources of uncertainty. When the extrapolated volumes are included, the NRMSE is only slightly higher (7.4 %), but still acceptable for most lakes. However, for $h$ or $A$ observations that are much smaller or larger than the observed $h$-$A$



pairs in the regression, the extrapolation errors can be much higher (e.g. Roseires Reservoir). In general, relatively few $V_{GSW}$ or $V_{Altimetry}$ estimates use this extrapolation.

Figure 8 shows the relationship between satellite and in situ volume variations for Lake Mead and illustrates the accuracy of the methodology. The estimates show strong linearity with in situ data, as shown by the correlation coefficient of > 0.99

5   for both $V_{GSW}$ as $V_{Altimetry}$. Both estimates are very close to the 1:1 line and therefore have a low NRMSE of 5.43 and 1.87 for $V_{GSW}$ and $V_{Altimetry}$ respectively. The NRMSE of $V_{GSW}$ is slightly higher, as this includes the extrapolated volume estimations from volume variations of 2 to -8 km$^3$. These volumes were estimated using area observations outside the range of $h$-$A$ pairs used in the regression. It is clear that the hypsometry of the lake does not hold perfectly true for these calculated areas.

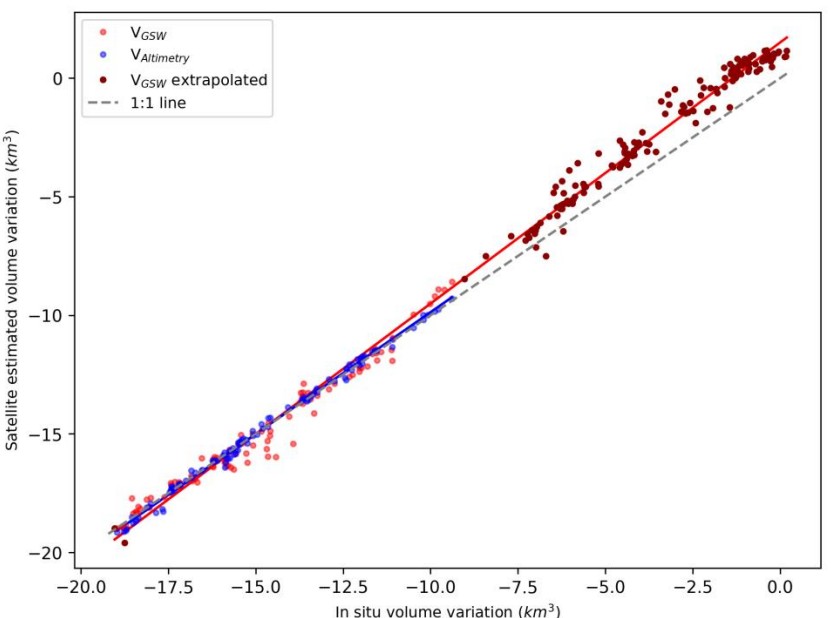

**Figure 8 Relationship in volume variation between satellite estimated and in situ observations for Lake Mead, with a Pearsons *r* > 0.99 and a NRMSE of 5.43 % ($V_{GSW}$) *and* 1.87 % ($V_{Altimetry}$).**

Figure 10 and Figure 10 show the validation volume time series against the satellite estimated volumes for Lake Mead and Lake Powell respectively. The time series show that both the timing and magnitude of the estimated volume fluctuations are

15   accurate for these lakes. For the non-extrapolated part of Lake Mead (2002-2016), estimated volumes were almost equal to the validation data. The extrapolated part (before 2002) was slightly overestimated. However, the dynamics over this period are still well captured. Lake Powell also showed accurate results, for both the seasonal and inter-annual fluctuations in water storage. Noteworthy for both lakes is the density of area observations due to a low amount of no data in the GSW dataset. For Lake Powell, the $V_{GSW}$ even correctly captured seasonal fluctuations, which is not always the case (see discussion). The

20   high accuracy is expressed in a low NRMSE of 4.52 and 2.96 % for $V_{GSW}$ and $V_{Altimetry}$ respectively. Note that almost no extrapolated volume estimates were required for the volume time series of Lake Powell.



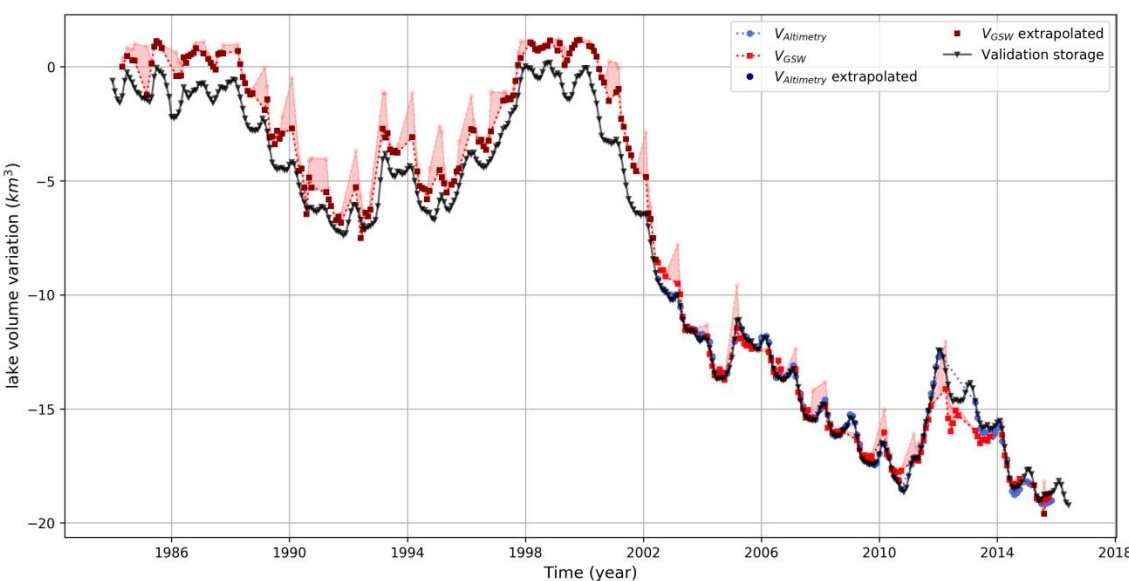

**Figure 9 Validation time series plotted with estimated reservoir volumes for Lake Mead. The black triangle line represents the validation storage as measured using the full lake bathymetry.**

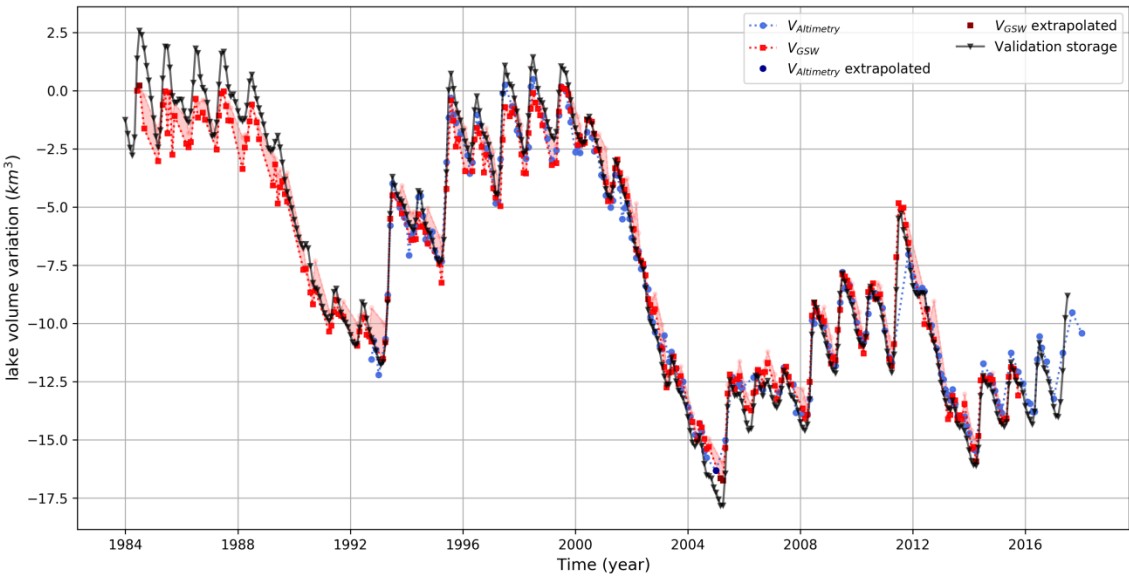

5   **Figure 10 Validation time series plotted with estimated reservoir volumes for Lake Powell. The black triangle line represents the validation storage as measured using the full lake bathymetry.**

**5. Discussion**

This study presented a new methodology to estimate lake and reservoir volumes using remote sensing alone. The validation showed that the method can produce water storage change estimates for many lakes, thus highlighting the



potential of combining satellite altimetry and the GSW dataset to develop a global lake and reservoir volume variation dataset. The GSW dataset global coverage, 30 m resolution, high accuracy and monthly surface water observations over a 32-year period increases the number of analysed lakes and the accuracy, quantity and temporal range of lake area calculations. Therefore, volume variations can now also be calculated using GSW lake areas as input independent from

altimetry data, which allows for volume calculations further back in time to 1984. The lake volume variations estimated here are likely to be useful in unravelling the causes of changing water availability due to climate change, extreme climate phenomena such as El Niño and La Niña, human abstractions and reservoir construction and operations. The volume estimations can help to validate large scale hydrological models and thus to improve future projections. Also, understanding volumetric lake and reservoir changes will be beneficial for regional and global flood and drought forecasting systems.

Finally, hydropower estimations may also be improved using these data.

    The methodology used in this study has a couple of limitations. They arise mainly from limitations in the input data (GSW and DAHITI altimetry dataset) and the volume calculation methodology.

### 5.1 Satellite altimetry limitations

    The altimeter footprint can be up to 16 m over land and land influences inside this footprint can alter the water level

accuracy by disturbing the altimeter waveforms (Fu and Cazenave, 2001). Water level estimations of very small lakes or reservoirs can therefore have a RMSE of several decimeters or larger (Schwatke et al., 2015a). We analysed many small lakes in Europe that did not show accurate regressions when they were relatively small (e.g. Thülsfelder Talsperre, 1.7 km$^2$, Hainer See, 4 km$^2$, Lake Resia, 6.6 km$^2$ and Altmuehl See, 4.5 km$^2$). Moreover, water level estimations are only possible if the water body is located along mission-dependent ground tracks. Larger lakes or reservoirs therefore have a higher

probability of frequent observations and a higher accuracy than smaller inland water bodies.

### 5.2 The influence of no data in lake area calculations

    The classifier of the GSW dataset has been shown to be very accurate, with less than 1 % of false water detections, and less than 5 % omission (Pekel et al., 2016). Therefore, the influence of classification errors in the volume estimations was very limited.

In this methodology, no data classifications in the GSW dataset play a much more important role. These are caused by snow, ice, cloud or sensor-related issues (e.g. white striping in Figure 2) and are likely to give an underestimation of the actual lake area if they are inside the mwe. Therefore, their influence has been reduced with strict no data thresholds applied to each monthly calculation. The 1% no data threshold for the regression and 5 % for the volume calculation resulted in the best trade-off between the number of observations from the GSW dataset and the accuracy of the estimates. Higher no data

thresholds introduced too much variability in the volume estimates, as (1) regression lines became much noisier and (2) the area of monthly no data exceeded the actual lake area variation. Locations with frequent cloud cover, lake ice coverage or sensor-related image failures will often return no data amounts that exceed the threshold, resulting in a sparse $V_{GSW}$ time





series. Although they are sparse, these area volume estimates can still show an acceptable accuracy, as was observed for the Puente Nuevo Reservoir in Spain (Figure 11). For locations in the high or low latitudes, winter months are masked due to low solar zenith angles that cause considerable shadowing. Moreover, a short period of daylight or ice and snow coverage on the water surface can further increase the amount of no data in these regions.

Besides the varying image quality, the whole stack of historical Landsat imagery has an unequal global spatial distribution. The U.S. and Australia are well covered, while other regions, like Africa and Southwestern Europe, have much less Landsat observations (Pekel et al., 2016; Wulder et al., 2016). The volumetric plots of U.S. lakes therefore typically show a highly dense $V_{\text{GSW}}$, while lakes in Africa and other less monitored regions typically have more years of no observations, especially before 2000. This can clearly be observed by comparing Lake Mead and Powell (U.S.) with Lake Nasser and Kariba (Africa)

in Figure 6. However, for lakes with sparse $V_{\text{GSW}}$ estimates, $V_{\text{Altimetry}}$ estimates still provide a valuable volume variation record.

Many layers in the GSW dataset could be used to reduce the amount of no data pixels. This would considerably increase the capabilities of the methodology, especially in the regions mentioned above. No data pixels that are located in the middle of the lake and are surrounded by water pixels (e.g. Figure 2) have a high probability of being water. Furthermore, the large

temporal range (1984-2015) of the GSW dataset could be used to further decrease the amount of no data. For example, no data pixels that are classified as water over nearly all the 380 months could be assigned to the water class with high confidence. The GSW seasonality layer could also be used to find permanent water pixels, which can be converted to water if they are not observed during a specific month.

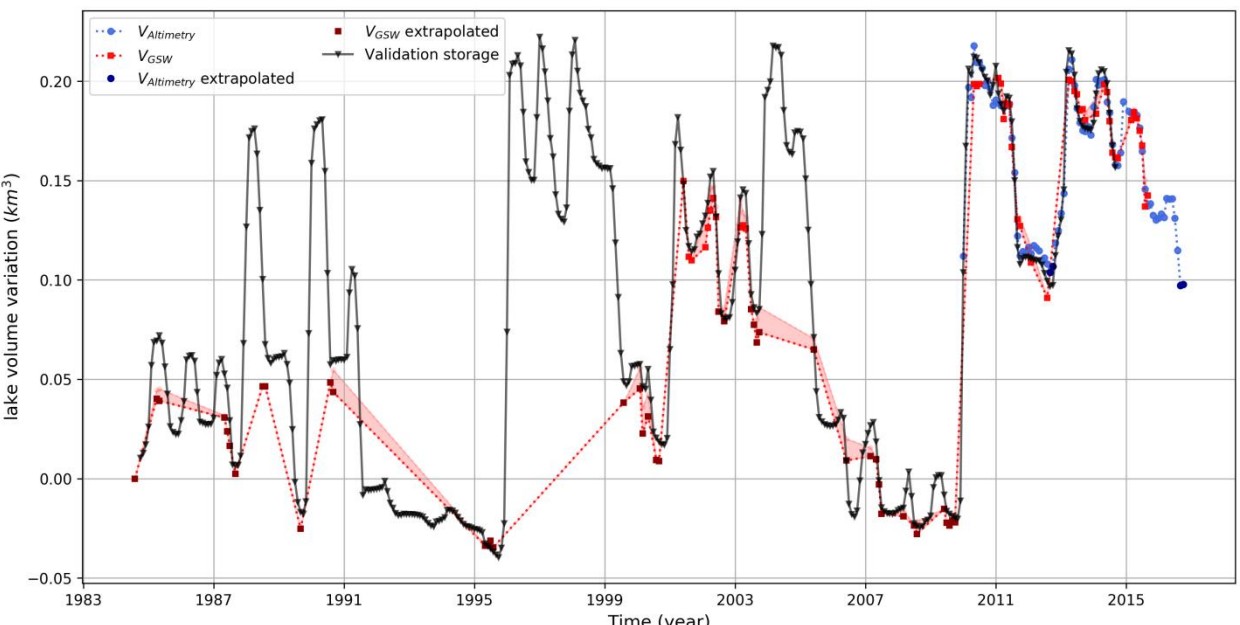

**Figure 11 Time series of satellite estimated volume variations compared to validation storages for Puente Nuevo Reservoir.**





### 5.3 Uncertainties and limitations in the volume calculation technique

Only three out of 135 lakes (Tawakoni, Urmia and Eagle) gave a clear non-linear hypsometry relationship and for these lakes volumes are assumed to be unreliable. Their regressions could be explained by a second- or third-order polynomial, as shown for Lake Urmia in Figure 12. This non-linearity is caused by a considerable change in slope, which will mostly be observed for lakes with extremely low water levels (e.g. Lake Urmia) or during floods.

To reduce data size, the monthly GSW dataset does not include exact dates of the Landsat observations. This causes more uncertainty in the regression, as the level and area observations may refer to different dates and therefore to slightly different lake conditions. For lakes with a highly variable level within a month, this uncertainty therefore increases.

For 35 lakes with lower performance, the regression showed relatively low $R^2$ values with an average of 0.52. The most important reasons for these bad regressions were found to be winter ice or snow coverage, small lake sizes, cloud coverage, Landsat sensor issues and multiple individual lake compartments. Winter ice coverage influences both the altimetry accuracy and the accuracy in the GSW water classification (e.g. Lake Ulungur, Rybinsk Reservoir, Reindeer Lake and Lake Ilmen). The current methodology needs to be refined to include lakes that split into multiple lakes during extreme drying (e.g. Aral Sea). As the different compartments can have different water level dynamics, individual regressions would need to be assigned to each compartment to calculate volume fluctuations of each compartment individually. Further research on water losses in the Aral Sea using this technique could be very promising.

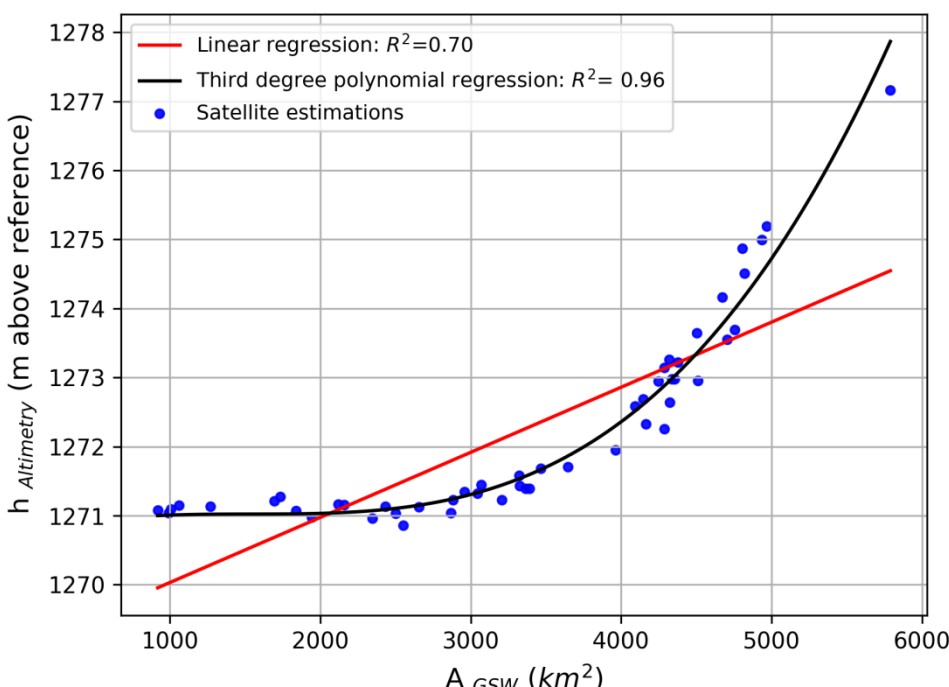

**Figure 12 Relationship between $A_{\text{GSW}}$ and $h_{\text{Altimetry}}$ for Lake Urmia. Only 3 out of the 135 lakes showed a clear non-linear relationship.**





The calculated extrapolated volumes are more uncertain than the non-extrapolated ones, depending on whether the hypsometry relationship holds true or changes for the observed $h$ or $A$ value outside the regression range. For lakes formed in V-shaped river valleys with deep and straight slopes (e.g. Lake Mead), the hypsometry relationship is likely to hold for extremely low or high water levels. However, if the altimetry measurement period is short, the range of values captured by

the regression line is likely to be small and the volume extrapolation using extreme level or area observations is expected to be inaccurate. This was observed for the Roseires Reservoir. However, including extrapolated volumes hardly changed the overall validation results, because the amount of total extrapolated volumes compared to non-extrapolated volumes is very small.

## 6. Conclusions and perspectives

This study successfully combined the JRC Global Surface Water (GSW) dataset and the DAHITI satellite altimetry dataset to estimate lake and reservoir volume fluctuations over all continents. The GSW dataset records surface water over a 32-year period, containing 3066080 monthly images that cover 99.95 % of the landmass. The extensive size and high accuracy of this surface water dataset allowed for detailed volume variation estimations over a very long time period (1984-2015), without being constrained by complex and computationally intensive classification procedures.

Lake areas from the GSW dataset and water levels from the DAHITI altimetry dataset have been combined in a regression to explain the lake hypsometry of 135 lakes globally. Nearly all lakes showed a linear regression. 57 of these lakes showed a highly accurate regression with $R^2 > 0.8$, with an average of 0.91. Lake volumes were calculated for all these lakes and for 41 other lakes with a nearly constant lake area. For 35 lakes, regressions were less accurate with an average $R^2$ of 0.52. Winter ice coverage, small lake size and no data in the GSW dataset were found to be most important reasons for low $R^2$ values.

For 98 lakes (57 variable area and 41 nearly constant area) volume variations were calculated by integrating the hypsometric relationships, using both area and water level observations separately. Decreases in water storage were found in Western U.S., where Lake Mead, Powell and the Great Salt Lake lost respectively 10 $km^3$, 6 $km^3$ and 16 $km^3$ of average volume between 1984-2000 and 2000-2015. According to our estimates, Lake Mead lost approximately 20 $km^3$ of water from 2000 to 2015. Using the USGS in situ measured storage volume of 31 $km^3$ in 2000, this is an almost 70 % reduction of

water storage over last 15 years. Lake volume variation estimates have been validated for 18 lakes in the U.S., Spain, Australia and Africa using in situ lake volume time series. The estimated volume variations showed the method to be very accurate, expressed in an average Pearson correlation coefficient of 0.97, and a normalized RMSE of 7.42 %.

The low number of adequate Landsat lake area observations for some regions like Africa and Southwestern Europe still remains a limitation. Therefore, it would be highly beneficial for the purpose of this research to include surface water data

from other satellites in the GSW dataset and to develop techniques to decrease the amount of no data in the current dataset. Future plans are to include Sentinel-1 and Sentinel-2 in the GSW dataset. The DAHITI database is continuously growing by analysing new water bodies, and newly available altimetry data will be processed to expand the volumetric dataset.



The lake volume variations should be analysed to unravel the contributions of climate change, extreme climate phenomena (like El Niño and La Niña), human abstractions and dam regulations on water storage changes. It is crucial to fully understand lake dynamics to improve future water resources projections in hydrological models and flood and drought forecasting systems.

5    This study constitutes a proof of concept paving the way for increasing the number of lakes and reservoirs analysed, which could potentially be included as an a priori water storage dataset for the Surface Water and Ocean Topography (SWOT) hydrology and oceanography satellite mission. Launched in 2020, this mission will combine water body contours and accurate water level estimations to estimate storage changes in lakes and reservoirs with an average accuracy of 20 cm (Biancamaria et al., 2016; Crétaux et al., 2015). The SWOT satellite will be unique due to its accuracy and capabilities on
10   smaller water bodies with a size of at least 250 x 250 m.

*Data availability.* Underlying research data are not publicly accessible. Lakes and reservoir volumes may become publicly available in a later stage through the DAHITI web service (http://dahiti.dgfi.tum.de/en/). The access to remote sensing data used in this study is explained in Sect. 2.





## Appendix 1

### Constant area lakes (Lc) overview

| Lake/ Reservoir name | Latitude | Longitude | min area (km$^2$) | max area (km$^2$) | CV area | min water level (m) | max water level (m) | R2 regression | Average volume change between 2000-2008 and 2008-2015 (km$^3$) |
|---|---|---|---|---|---|---|---|---|---|
| Albert | 1.63 | 30.91 | 5354.80 | 5412.16 | 0.003 | 620.71 | 622.55 | 0.01 | -3.49 |
| Argentino | -50.24 | -72.84 | 1546.95 | 1555.79 | 0.004 | 177.04 | 180.78 | 0.01 | 0.40 |
| Athabasca | 59.19 | -109.28 | 7528.69 | 7723.01 | 0.005 | 208.71 | 211.31 | 0.53 | -0.59 |
| Baikal | 53.36 | 107.57 | 31572.61 | 31963.98 | 0.003 | 455.00 | 455.95 | 0.00 | 2.54 |
| Baker | 64.16 | -95.28 | 1697.53 | 1723.77 | 0.004 | 0.62 | 2.14 | 0.45 | 0.44 |
| Balaton | 46.86 | 17.75 | 573.48 | 582.15 | 0.003 | 103.98 | 105.24 | 0.03 | - |
| Buenos Aires | -46.55 | -71.97 | 1832.01 | 1862.79 | 0.003 | 205.25 | 207.13 | 0.23 | 0.13 |
| Chiemsee | 47.88 | 12.45 | 74.89 | 76.50 | 0.005 | 517.86 | 519.55 | 0.27 | -0.01 |
| Constance | 47.61 | 9.42 | 466.60 | 469.81 | 0.002 | 394.09 | 396.01 | 0.22 | 0.12 |
| Dore | 54.77 | -107.31 | 621.17 | 632.80 | 0.004 | 458.79 | 459.94 | 0.49 | 0.28 |
| Edward | -0.36 | 29.59 | 2209.06 | 2235.13 | 0.003 | 913.98 | 915.54 | 0.37 | 0.52 |
| Erie | 42.14 | -81.29 | 25488.04 | 25789.63 | 0.003 | 173.55 | 174.87 | 0.00 | 2.36 |
| Flathead | 47.88 | -114.14 | 480.89 | 494.76 | 0.006 | 878.96 | 882.12 | 0.40 | -0.13 |
| Great Bear | 66.00 | -120.97 | 30266.01 | 30555.78 | 0.003 | 156.95 | 157.75 | 0.17 | 4.49 |
| Huron | 45.01 | -82.29 | 58972.33 | 59664.21 | 0.003 | 175.32 | 176.97 | 0.10 | 8.16 |
| Issyk-Kul | 42.44 | 77.27 | 6165.81 | 6220.68 | 0.002 | 1605.55 | 1606.32 | 0.04 | 0.16 |
| Keller | 63.93 | -121.58 | 389.56 | 393.02 | 0.002 | 239.69 | 240.77 | 0.07 | 0.06 |
| Khuvsgul | 51.06 | 100.47 | 2758.95 | 2788.74 | 0.002 | 1646.49 | 1647.39 | 0.03 | 0.56 |
| Kivu | -2.04 | 29.10 | 2368.70 | 2395.02 | 0.003 | 1460.74 | 1462.34 | 0.00 | 0.56 |
| Kremenchuk | 49.30 | 32.68 | 1828.27 | 1914.02 | 0.006 | 76.70 | 81.57 | 0.32 | 0.35 |
| Ladoga | 60.84 | 31.47 | 17330.63 | 17601.68 | 0.003 | 2.27 | 5.11 | 0.03 | 8.10 |
| Llanquihue | -41.14 | -72.82 | 855.85 | 864.19 | 0.002 | 50.00 | 51.00 | 0.00 | -0.08 |
| Malawi | -12.02 | 34.53 | 29195.33 | 29550.33 | 0.003 | 472.15 | 475.22 | 0.03 | -4.18 |
| Michigan | 44.02 | -86.76 | 57283.26 | 57856.91 | 0.002 | 175.25 | 176.91 | 0.00 | 7.59 |
| Mweru | -9.02 | 28.72 | 4963.59 | 5104.84 | 0.006 | 923.20 | 926.72 | 0.01 | 2.24 |
| Nicaragua | 11.53 | -85.41 | 7749.76 | 7815.10 | 0.003 | 30.05 | 32.56 | 0.02 | 1.22 |
| Nipigon | 49.81 | -88.52 | 4462.91 | 4515.99 | 0.003 | 259.49 | 260.91 | 0.00 | 1.24 |
| Novosibirsk | 54.54 | 82.37 | 976.27 | 998.96 | 0.005 | 111.69 | 114.47 | 0.00 | -0.02 |
| Ontario | 43.64 | -77.81 | 18529.57 | 18741.92 | 0.003 | 73.97 | 75.37 | 0.04 | 0.37 |
| Peipus | 58.54 | 27.55 | 3465.82 | 3556.71 | 0.005 | 28.78 | 31.13 | 0.18 | 0.62 |
| Porisvatn | 64.27 | -18.86 | 84.49 | 85.26 | 0.003 | 570.71 | 580.61 | 0.60 | -0.06 |





| | | | | | | | | | |
|---|---|---|---|---|---|---|---|---|---|
| **Ranco** | -40.24 | -72.40 | 424.58 | 429.46 | 0.003 | 63.10 | 65.96 | 0.00 | 0.00 |
| **Saint Jean** | 48.59 | -72.04 | 1063.55 | 1086.20 | 0.004 | 97.74 | 101.87 | 0.07 | 0.06 |
| **Shala** | 7.47 | 38.51 | 299.65 | 308.01 | 0.006 | 1553.45 | 1555.58 | 0.38 | -0.14 |
| **Superior** | 47.55 | -87.78 | 81320.65 | 82151.16 | 0.003 | 182.32 | 183.43 | 0.02 | 5.31 |
| **Tanganyika** | -6.22 | 29.89 | 32400.06 | 32685.75 | 0.003 | 768.43 | 770.94 | 0.03 | 14.04 |
| **Uvs** | 50.32 | 92.75 | 3486.72 | 3628.63 | 0.008 | 761.81 | 763.25 | 0.57 | -1.26 |
| **Vanern** | 58.90 | 13.27 | 5300.96 | 5414.91 | 0.005 | 44.22 | 45.33 | 0.12 | 0.95 |
| **Victoria** | -1.12 | 32.90 | 66123.35 | 66765.69 | 0.004 | 1133.55 | 1135.95 | 0.07 | 11.55 |
| **Wollaston** | 58.23 | -103.29 | 2192.42 | 2227.67 | 0.004 | 395.51 | 396.90 | 0.03 | 0.01 |
| **Yellowstone** | 44.43 | -110.37 | 338.88 | 343.35 | 0.003 | 2357.03 | 2358.79 | 0.45 | 0.10 |

## Variable area lakes (LvG) overview

| Lake/reservoir name | Latitude | Longitude | min area (km$^2$) | max area (km$^2$) | CV area | min water level (m) | max water level (m) | R2 regression | Average volume change between 1984-2000 and 2000-2016 (km$^3$) |
|---|---|---|---|---|---|---|---|---|---|
| **Alcantara** | 39.77 | -6.67 | 43.22 | 63.72 | 0.098 | 187.83 | 216.33 | 0.88 | 0.43 |
| **Almanor** | 40.25 | -121.14 | 87.07 | 100.89 | 0.028 | 1367.71 | 1373.24 | 0.85 | 0.03 |
| **Alvaro Obregon** | 27.96 | -109.86 | 34.71 | 177.09 | 0.394 | 64.51 | 93.03 | 0.91 | -0.29 |
| **Angostura** | 16.12 | -92.64 | 265.26 | 562.21 | 0.143 | 496.84 | 527.92 | 0.98 | -1.71 |
| **Argyle** | -16.34 | 128.75 | 346.54 | 1187.54 | 0.195 | 88.66 | 95.83 | 0.89 | 3.30 |
| **Assad** | 36.00 | 38.26 | 518.94 | 643.55 | 0.044 | 297.60 | 304.36 | 0.85 | 0.52 |
| **Bagre** | 11.56 | -0.68 | 0.00 | 195.02 | 0.555 | 227.12 | 236.06 | 0.98 | 0.44 |
| **Balbina** | -1.44 | -59.89 | 23.36 | 2421.59 | 0.315 | 42.12 | 49.09 | 0.92 | 5.51 |
| **Barragem do Caia** | 39.03 | -7.19 | 5.54 | 15.72 | 0.197 | 222.06 | 233.22 | 0.97 | 0.01 |
| **Berryessa** | 38.58 | -122.22 | 45.30 | 72.77 | 0.101 | 120.67 | 134.76 | 0.95 | 0.28 |
| **Boston** | 41.97 | 87.06 | 901.79 | 1063.23 | 0.052 | 1045.34 | 1049.88 | 0.98 | 0.31 |
| **Brahmamsagar** | 14.78 | 78.89 | 0.09 | 21.30 | 0.930 | 186.79 | 209.36 | 0.99 | 0.13 |
| **Bratsk** | 55.87 | 102.34 | 2908.56 | 3126.58 | 0.019 | 395.21 | 402.24 | 0.88 | - |
| **Cahora Bassa** | -15.68 | 31.67 | 1629.92 | 2481.94 | 0.128 | 318.20 | 327.01 | 0.94 | 19.38 |
| **Chapala** | 20.24 | -103.02 | 717.29 | 1102.63 | 0.107 | 1517.52 | 1522.95 | 0.82 | -0.08 |
| **Chiquita** | -30.54 | -62.66 | 2866.43 | 6763.70 | 0.204 | 67.41 | 72.51 | 0.84 | -0.48 |
| **Encoro de Salas** | 41.92 | -7.93 | 1.06 | 3.75 | 0.191 | 815.95 | 828.93 | 0.95 | 0.01 |
| **Eucumbene** | -36.08 | 148.70 | 53.47 | 131.58 | 0.210 | 1117.82 | 1155.38 | 0.97 | -1.35 |
| **Great Salt Lake** | 41.18 | -112.53 | 3228.40 | 6205.50 | 0.182 | 1278.83 | 1283.74 | 0.93 | -15.72 |
| **Guri** | 7.40 | -62.86 | 2914.13 | 3507.38 | 0.076 | 243.73 | 271.17 | 0.99 | -7.26 |
| **Houston** | 29.98 | -95.14 | 31.19 | 38.24 | 0.031 | 11.34 | 13.70 | 0.87 | 0.00 |



| | | | | | | | | |
|---|---|---|---|---|---|---|---|---|
| **Hubbard Creek** | 32.79 | -99.01 | 15.28 | 58.58 | 0.281 | 351.42 | 360.75 | 0.98 | -0.14 |
| **Hulun** | 48.95 | 117.40 | 1782.96 | 2122.60 | 0.050 | 540.45 | 544.01 | 0.97 | -3.01 |
| **Kainji** | 10.32 | 4.56 | 724.06 | 1134.86 | 0.075 | 128.60 | 139.71 | 0.93 | -0.48 |
| **Kajaki** | 32.33 | 65.19 | 23.08 | 38.44 | 0.088 | 998.19 | 1021.17 | 0.88 | -0.03 |
| **Kapchagay** | 43.82 | 77.61 | 1119.35 | 1242.98 | 0.021 | 475.34 | 479.41 | 0.94 | 0.17 |
| **Karakaya Baraji** | 38.53 | 38.47 | 21.41 | 236.14 | 0.141 | 673.61 | 692.12 | 0.97 | 0.81 |
| **Kariba** | -16.99 | 28.04 | 4571.10 | 5323.91 | 0.041 | 475.54 | 487.09 | 0.96 | 15.28 |
| **Khyargas Nuur** | 49.18 | 93.32 | 1364.23 | 1397.40 | 0.008 | 1027.99 | 1032.76 | 0.97 | 1.49 |
| **Manitoba** | 50.98 | -98.70 | 4700.32 | 5022.90 | 0.012 | 246.51 | 248.63 | 0.82 | 0.47 |
| **Manso** | -14.95 | -55.66 | 0.24 | 339.87 | 0.847 | 283.03 | 287.27 | 0.83 | 4.72 |
| **Massinger Barragen** | -23.89 | 32.04 | 16.25 | 128.93 | 0.342 | 100.50 | 123.66 | 0.93 | 1.10 |
| **Mead** | 36.19 | -114.41 | 327.92 | 580.50 | 0.181 | 328.69 | 353.61 | 0.98 | -10.75 |
| **Nasser** | 22.88 | 32.33 | 3156.39 | 5770.06 | 0.128 | 166.85 | 181.08 | 0.92 | -1.79 |
| **Netzahualcoyotl** | 17.14 | -93.63 | 228.21 | 281.65 | 0.053 | 151.88 | 177.34 | 0.94 | -1.12 |
| **O. H. Ivie** | 31.55 | -99.72 | 0.10 | 70.45 | 0.642 | 458.64 | 469.13 | 0.97 | -0.15 |
| **Powell** | 37.24 | -110.96 | 311.38 | 555.09 | 0.169 | 1086.86 | 1126.50 | 0.99 | -6.35 |
| **Puente Nuevo** | 38.13 | -4.97 | 3.44 | 17.87 | 0.401 | 437.24 | 444.78 | 0.91 | 0.10 |
| **Qarun** | 29.47 | 30.63 | 226.46 | 255.33 | 0.032 | -42.59 | -41.40 | 0.86 | 0.12 |
| **Qinghai** | 36.89 | 100.20 | 4217.58 | 4369.47 | 0.012 | 3193.29 | 3194.35 | 0.83 | -0.36 |
| **Richland Chambers** | 31.99 | -96.26 | 1.32 | 174.31 | 0.372 | 93.08 | 96.64 | 0.92 | 0.27 |
| **Roseires** | 11.60 | 34.46 | 88.65 | 578.54 | 0.436 | 471.70 | 487.74 | 0.84 | 2.06 |
| **Rukwa** | -7.96 | 32.21 | 5388.63 | 6057.51 | 0.039 | 799.90 | 804.28 | 0.88 | -13.01 |
| **Sam Rayburn** | 31.22 | -94.24 | 290.73 | 429.91 | 0.067 | 46.35 | 53.02 | 0.88 | 0.00 |
| **Sarygamysh** | 41.93 | 57.41 | 3086.11 | 3967.24 | 0.065 | 0.53 | 8.49 | 0.99 | 20.50 |
| **Serena** | 38.89 | -5.18 | 3.18 | 125.32 | 0.386 | 330.93 | 349.84 | 0.96 | 1.69 |
| **Sobradinho** | -9.67 | -41.62 | 1501.04 | 3446.70 | 0.216 | 384.92 | 394.58 | 0.86 | -2.20 |
| **Tengiz** | 50.44 | 69.08 | 861.36 | 1632.89 | 0.168 | 304.32 | 306.60 | 0.95 | -1.25 |
| **Tharthar** | 34.01 | 43.26 | 1593.83 | 2286.33 | 0.119 | 42.76 | 63.60 | 0.94 | -16.29 |
| **Titicaca** | -15.90 | -69.33 | 7513.44 | 8361.62 | 0.023 | 3809.10 | 3811.66 | 0.81 | -2.60 |
| **Toktogul** | 41.79 | 72.90 | 195.07 | 289.14 | 0.117 | 856.13 | 898.94 | 0.91 | -0.73 |
| **Toledo Bend** | 31.47 | -93.72 | 472.87 | 630.83 | 0.056 | 48.86 | 52.65 | 0.81 | 0.01 |
| **Volta** | 7.44 | -0.18 | 4645.90 | 6885.92 | 0.124 | 74.54 | 84.85 | 0.96 | -14.51 |
| **Walker** | 38.68 | -118.71 | 111.70 | 154.99 | 0.081 | 1191.20 | 1205.93 | 0.98 | -0.96 |
| **Williston** | 56.08 | -123.66 | 1495.12 | 1715.51 | 0.031 | 656.92 | 671.72 | 0.84 | 2.72 |
| **Winnipegosis** | 52.55 | -100.15 | 5027.12 | 5257.38 | 0.008 | 252.43 | 254.72 | 0.87 | 2.78 |
| **Yesa** | 42.61 | -1.11 | 11.50 | 17.42 | 0.086 | 459.12 | 486.80 | 0.86 | 0.02 |





**Less performant variable area lakes (LvP) overview**

| Lake/reservoir name | Latitude | Longitude | min area (km²) | max area (km²) | CV area | min water level (m) | max water level (m) | R2 regression |
|---|---|---|---|---|---|---|---|---|
| Altmuhl | 49.13 | 10.72 | 3.28 | 3.44 | 0.012 | 414.00 | 416.01 | 0.19 |
| Aydar | 40.87 | 66.91 | 1661.19 | 3228.23 | 0.174 | 244.61 | 248.04 | 0.71 |
| Bardwell | 32.28 | -96.66 | 10.72 | 14.48 | 0.046 | 127.03 | 131.91 | 0.55 |
| Beysehir | 37.78 | 31.51 | 611.43 | 669.54 | 0.020 | 1120.43 | 1124.22 | 0.73 |
| Caddabassa | 8.87 | 39.87 | 30.15 | 48.89 | 0.141 | 560.51 | 563.13 | 0.11 |
| Caddo | 32.71 | -94.02 | 40.75 | 63.30 | 0.125 | 50.59 | 52.87 | 0.44 |
| Cedar | 53.34 | -100.17 | 2358.60 | 2709.49 | 0.025 | 252.59 | 256.56 | 0.45 |
| Chad | 13.04 | 14.49 | 1241.46 | 1459.73 | 0.044 | 279.96 | 282.13 | 0.30 |
| Chamo | 5.85 | 37.55 | 292.56 | 330.97 | 0.030 | 1105.26 | 1108.24 | 0.66 |
| Churumuco | 18.56 | -101.86 | 156.26 | 309.69 | 0.157 | 137.28 | 161.38 | 0.51 |
| Claire | 58.59 | -112.09 | 1231.20 | 1416.09 | 0.036 | 209.60 | 210.73 | 0.21 |
| Danau Tang | 0.63 | 112.47 | 2.81 | 6.50 | 0.170 | 17.77 | 24.78 | 0.62 |
| Eagle | 40.65 | -120.73 | 51.08 | 102.63 | 0.093 | 1551.70 | 1556.77 | 0.75 |
| Fairfield | 31.79 | -96.06 | 6.12 | 8.03 | 0.059 | 92.84 | 95.17 | 0.26 |
| Hainer | 51.17 | 12.46 | 0.00 | 3.70 | 0.586 | 123.52 | 127.82 | 0.00 |
| Ilmen | 58.25 | 31.38 | 941.97 | 1306.72 | 0.112 | 15.84 | 21.45 | 0.62 |
| Kuybyshev | 54.60 | 49.17 | 4529.53 | 4839.51 | 0.012 | 47.73 | 53.17 | 0.61 |
| Lesser Slave | 55.44 | -115.40 | 1120.94 | 1182.22 | 0.009 | 576.09 | 578.14 | 0.60 |
| Mosul Dam | 36.74 | 42.75 | 12.90 | 331.67 | 0.224 | 303.70 | 329.06 | 0.73 |
| Musters | -45.40 | -69.20 | 422.71 | 465.35 | 0.025 | 268.54 | 271.00 | 0.78 |
| Poopo | -18.76 | -67.09 | 10.62 | 3104.92 | 0.583 | 3685.50 | 3686.24 | 0.21 |
| Reindeer | 57.28 | -102.38 | 5201.41 | 5422.67 | 0.010 | 335.58 | 337.42 | 0.67 |
| Resia | 46.80 | 10.53 | 5.77 | 6.12 | 0.015 | 1476.00 | 1496.18 | 0.76 |
| Rybinsk | 58.52 | 38.28 | 3590.26 | 3778.11 | 0.012 | 97.96 | 101.16 | 0.53 |
| Tai Hu | 31.20 | 120.23 | 2216.78 | 2359.42 | 0.016 | 1.36 | 2.67 | 0.00 |
| Tana | 11.99 | 37.31 | 2986.63 | 3107.77 | 0.008 | 1784.94 | 1788.29 | 0.74 |
| Tawakoni | 32.89 | -96.00 | 106.85 | 150.45 | 0.070 | 131.01 | 134.08 | 0.76 |
| Thulsfelder | 52.93 | 7.93 | 0.42 | 1.18 | 0.162 | 21.61 | 23.03 | 0.63 |
| Tsimlyansk | 47.96 | 42.79 | 1988.14 | 2366.59 | 0.031 | 31.70 | 36.38 | 0.80 |
| Turkana | 3.51 | 36.20 | 7033.64 | 7501.42 | 0.015 | 360.37 | 365.18 | 0.71 |
| Ulungur | 47.25 | 87.29 | 852.58 | 887.48 | 0.009 | 482.36 | 484.04 | 0.24 |
| Urmia | 37.64 | 45.50 | 917.87 | 5790.55 | 0.372 | 1270.11 | 1278.01 | 0.70 |
| Zama | 58.79 | -119.03 | 65.19 | 288.52 | 0.443 | 325.88 | 328.81 | 0.28 |
| Zaysan | 48.01 | 83.89 | 2829.26 | 3253.41 | 0.032 | 388.60 | 394.87 | 0.56 |




*Author contribution.* T. B. developed the methodology, the scripts and was the main writer of the paper. A.R. helped writing the introduction section and gave comprehensive scientific support over the whole period. E.G. considerably supported in designing the methodology and in coding the scripts. C.S. provided DAHITI altimetry data, additionally processed water bodies on demand and moreover wrote the altimetry section. J.F.P. and A.C. contributed in the Google Earth Engine usage

5    and wrote the GSW section. All authors actively contributed to the feedback process after the first draft.

*Competing interests.* The authors declare that they have no conflict of interest.

*Acknowledgements.* We would like to thank the Google Earth Engine team, and especially Matthew Hancher and Noel

10   Gorelick, for support in developing the Earth Engine scripts. We would also like to express our gratitude to Guido Lemoine (Joint Research Centre) for his technical support in Earth Engine and Steven de Jong (Utrecht University) for his feedback on the first report.



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

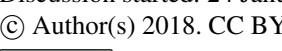



**Table 1 Overview of the validation results, excluding and including extrapolated volumes.**

| Lake/reservoir name: | | *Extrapolated volumes excluded* | | | *Extrapolated volumes included* | | |
|---|---|---|---|---|---|---|---|
| | | Pearsons *r* | RMSE (km³) | NRMSE (%) | Pearsons *r* | RMSE (km³) | NRMSE (%) |
| **Alcantara** | GSW dataset: | 0.941 | 0.229 | 12.750 | 0.941 | 0.229 | 12.750 |
| | Altimetry: | 0.986 | 0.151 | 8.427 | 0.987 | 0.153 | 8.265 |
| **Almanor** | GSW dataset: | 0.902 | 0.060 | 11.340 | 0.908 | 0.071 | 11.122 |
| | Altimetry: | 0.990 | 0.019 | 3.676 | 0.989 | 0.020 | 3.874 |
| **Argyle** | GSW dataset: | 0.954 | 0.471 | 7.924 | 0.954 | 0.471 | 7.924 |
| | Altimetry: | 0.995 | 0.147 | 2.474 | 0.995 | 0.147 | 2.474 |
| **Berryessa** | GSW dataset: | 0.961 | 0.063 | 7.448 | 0.989 | 0.076 | 5.311 |
| | Altimetry: | 0.978 | 0.043 | 5.575 | 0.978 | 0.043 | 5.575 |
| **Encoro de Salas** | GSW dataset: | 0.966 | 0.004 | 8.358 | 0.966 | 0.004 | 7.455 |
| | Altimetry: | 0.985 | 0.002 | 7.323 | 0.992 | 0.003 | 6.145 |
| **Eucumbene** | GSW dataset: | 0.933 | 0.114 | 11.802 | 0.933 | 0.114 | 11.802 |
| | Altimetry: | 0.964 | 0.084 | 8.419 | 0.964 | 0.084 | 8.419 |
| **Houston** | GSW dataset: | 0.904 | 0.008 | 10.235 | 0.904 | 0.008 | 10.235 |
| | Altimetry: | 0.938 | 0.005 | 5.666 | 0.938 | 0.005 | 5.436 |
| **Hubbard Creek** | GSW dataset: | 0.985 | 0.017 | 5.173 | 0.988 | 0.018 | 5.372 |
| | Altimetry: | 0.998 | 0.006 | 1.924 | 0.999 | 0.007 | 2.118 |
| **Mead** | GSW dataset: | 0.985 | 0.449 | 4.685 | 0.997 | 1.045 | 5.434 |
| | Altimetry: | 0.998 | 0.179 | 1.865 | 0.998 | 0.179 | 1.865 |
| **O. H. Ivie** | GSW dataset: | 0.985 | 0.019 | 4.924 | 0.993 | 0.030 | 4.950 |
| | Altimetry: | 0.999 | 0.006 | 1.784 | 0.999 | 0.006 | 1.784 |
| **Powell** | GSW dataset: | 0.993 | 0.913 | 4.947 | 0.994 | 0.923 | 4.524 |
| | Altimetry: | 0.997 | 0.552 | 3.074 | 0.998 | 0.554 | 2.964 |
| **Puente Nuevo** | GSW dataset: | 0.961 | 0.011 | 9.182 | 0.991 | 0.013 | 5.329 |
| | Altimetry: | 0.993 | 0.006 | 4.972 | 0.994 | 0.006 | 4.954 |
| **Richland Chambers** | GSW dataset: | 0.954 | 0.046 | 9.161 | 0.954 | 0.046 | 9.161 |
| | Altimetry: | 0.989 | 0.023 | 4.224 | 0.990 | 0.022 | 4.196 |
| **Roseires** | GSW dataset: | 0.805 | 0.295 | 18.872 | 0.971 | 2.434 | 41.455 |
| | Altimetry: | 0.925 | 0.214 | 12.443 | 0.962 | 0.214 | 11.473 |
| **Serena** | GSW dataset: | 0.989 | 0.092 | 4.907 | 0.989 | 0.122 | 3.767 |
| | Altimetry: | 0.990 | 0.077 | 4.294 | 0.990 | 0.077 | 4.294 |
| **Toledo Bend** | GSW dataset: | 0.765 | 0.351 | 14.376 | 0.777 | 0.345 | 14.135 |
| | Altimetry: | 0.988 | 0.122 | 5.015 | 0.988 | 0.121 | 4.974 |
| **Walker** | GSW dataset: | 0.936 | 0.071 | 10.798 | 0.971 | 0.072 | 8.180 |
| | Altimetry: | 0.988 | 0.032 | 4.882 | 0.990 | 0.041 | 4.499 |





| | | | | | | | |
|---|---|---|---|---|---|---|---|
| **Yesa reservoir** | GSW dataset: | 0.919 | 0.030 | 11.863 | 0.960 | 0.029 | 8.071 |
| | Altimetry: | 0.983 | 0.015 | 6.228 | 0.979 | 0.024 | 6.971 |
| **Average** | | **0.959** | **0.137** | **7.250** | **0.970** | **0.215** | **7.424** |