# Peer review of "A global lake and reservoir volume analysis using a surface water dataset and satellite altimetry"

_Hydrology and Earth System Sciences, 2018_

## Referee Comment (RC1) · Anonymous Referee #1 · 22 Mar 2018

**General comments**

This paper explores the use of the JRC global surface water dataset, and the DAHITI satellite altimetry database to estimate hypsometry relationships for a reasonable number of lakes across the globe. The paper should be of interest to a people working in water resources, and potentially is a publishable paper. At the moment however, the paper is a fairly simple data analysis with insufficient statistics (i.e. uncertainties) warrant publications as it is. Might improve the paper if the authors consider what they learnt from analysing the dataset, and what limits would they place on the size of the dam might suit using this approach, rather than given vague qualitative statements.

While the paper appears to be overall well written, there are some issues with the

material presented (see comments below), and it would be good to have estimates of the uncertainty in the regressed coefficients (maybe indicated through confidence bounds on the fitted functions shown in the plots would be best?). I think the paper needs some revision before being ready for publication.

**Specific comments**

1. Page 1, lines 19-25: the average r is given across 18 lakes. Would be good to know what the standard deviation is also as this would at least give the reader some idea of the scatter.

2. Page 4, lines 25-28: The definition of a large lake (ocean-like conditions) is a little vague. Might be useful to have a quantitative definition of what a large lake and a small lake are? Maybe something related to the minimum width of the widest part of the lake?

3. Page 10, Figure 4: there are large departures in the plot for Lake Nasser – what could cause these? How significant are they?

4. Page 11, lines 14: It would be good to give some information on how the uncertainty was obtained, and how the no data pixels were treated in estimating the points shown in Figure 6. Are the red points likely to be lower bounds on the lake volume? From the Figure, these seems to be the case. If they are lower bounds, then the red shaded area seems to span between this low bound and an estimated upper bound. How are the individual pixels within the MWE converted to an area? Is this simply adding up the number of pixels with a detection of a water surface? Appears to be so based on what I can see, in which case this is a very simplified approach, and better estimates of the upper and lower bounds could be made by considering the part of wet pixels at other times (I note that

some discussion on this appears in page 18, reinforcing my interpretation that the simplified approach has been used).

5. Page 14, lines 8-13: Information on data sources seems to be incomplete. Sources for US, Spain and Sudan are given, what about the source for the 2 lakes in Australia?

6. Page 14, lines 13-14: Seems strange to make the statement that the NRMSE is relatively low taking into account all sources of uncertainty, but there is no discussion about what the sources of uncertainty are that have been considered, or the magnitude of the overall uncertainty. Does this statement mean that the NRMSE is a lot smaller than would be expected given the estimated uncertainty? If so, it suggests a possible error in the uncertainty quantification (e.g. ignoring the impact of serial correlation between the different component uncertainties).

7. Page 15, line 1: Yes, extrapolating beyond the limits of the data will result in higher errors. This is why the uncertainty in the regressed coefficients should be reported. Even then, the uncertainty estimated from the regressed quantities will be a lower bound on the uncertainty in the extrapolation as the estimate is based on the assumption that the fitted function still holds. Possible explanation for the over-estimation of the extrapolated storage for Lake Mead shown in Figure 8 (regressed coefficients are time dependent, or relationship is not as linear as was originally thought), or is the red line shown there within the uncertainty bounds for the original regression?

8. Page 19, lines 2-5: A non-linear hypsometry relationship shouldn't mean the lake volumes are unreliable. Just that more care and some more maths is needed to derive the volumes. The main issue would be the choice of fitted function, and how this behaves under extrapolation. Given the result shown in Figure 12, a hyperbolic function that becomes roughly constant as area decreases, and linear

as area increases would likely be a much better function to fit than a quadratic or a cubic.

**Minor comments**

1. Page 5, line 26 (and elsewhere): might be better to have all acronyms in capital letters.

2. Page 9, line 11: "Lakes Powell, Kariba, Mead and Nasser"

3. Page 12, line 6 (and elsewhere): $km^3$ is not a standard SI unit. The equivalent SI unit would be TL (teralitres). Is $km^3$ acceptable?

4. Page 12, line 23: "during which time it lost approximately 30 $km^3$"?

---

## Referee Comment (RC2) · Anonymous Referee #2 · 29 May 2018

After reviewing the paper of lake volume analysis using satellite altimetry, it is found the topic is good and is of great practical interest, because the determination the relationship between volume and water level is very important for reservoir operation and for rational water allocation. The methods employed in the study are also good compared with the traditional methods by surveying. It will become popular for the estimation of lake volume in the remote region, such as mountain rgion. Therefore, it is recommended to publish in the journal.

---

## Referee Comment (RC3) · R. Romanowicz (Referee) · 8 Jun 2018

The paper presents an analysis of time series of variations of lake and reservoir volumes for 135 lakes from all over the world. The volumes are estimated using remote sensing alone. The JCR Global Surface Water (GSW) dataset with 30 m resolution and the DAHITI satellite altimetry dataset were combined to estimate lake and reservoir volume fluctuations over all continents.

It is an interesting paper that promotes the use of remote sensing data. It is also well written and well organised. The authors claim that their study can be useful in the assessment of changes in water availability due to climate change. However, I cannot see how the latter can be achieved based on the material presented. The authors are

asked to expand on that thought.

In addition, I am not sure how this publication might contribute to the enhancement of knowledge on the physical processes involved. In other words, a link is missing between the paper's main contribution – a new lake and reservoir volume dataset - and the possible applications.

The authors mention that some lakes showed poor regression between h and A, with one of the reasons being lake size. What are the limits of detection of h(A) relationship from the satellite data (what is the minimum lake area detectable)?

From the satellite altimetry data description we learn that the accuracy of water level time series varies with the lake size, from 4-5 cm for large lakes up to over a meter (several decimetres) for small lakes and rivers. Can this variable error variance be included in the calculation of volume variations?

---

## Author Comment (AC1) · 23 Jul 2018

*This paper explores the use of the JRC global surface water dataset, and the DAHITI satellite altimetry database to estimate hypsometry relationships for a reasonable number of lakes across the globe. The paper should be of interest to a people working in water resources, and potentially is a publishable paper.*

Thank you a lot for your time, effort and usefull feedback. We agree that our work should be of interest to many people working in water resources and hydrological modelling, as it improves on monitoring techniques of lakes and reservoirs currently available.

*At the moment however, the paper is a fairly simple data analysis with insufficient*

[Figure]

*statistics (i.e. uncertainties) warrant publications as it is. Might improve the paper if the authors consider what they learnt from analysing the dataset, and what limits would they place on the size of the dam might suit using this approach, rather than given vague qualitative statements. While the paper appears to be overall well written, there are some issues with the material presented (see comments below), and it would be good to have estimates of the uncertainty in the regressed coefficients (maybe indicated through confidence bounds on the fitted functions shown in the plots would be best?). I think the paper needs some revision before being ready for publication.*

We agree that we can improve on these factors, especially on the uncertainty analysis, the application of the dataset and the limitations of satellite altimetry. In the comments below, we try to clarify on these points by either improving current explanations or by extending the analysis. We propose to include these revisions in the revised paper. Also, we found a data gap in the Tibetan Plateau, so we would like to fill this gap by analyzing five additional lakes in this region.

**Specific comments**

1) *Page 1, lines 19-25: the average r is given across 18 lakes. Would be good to know what the standard deviation is also as this would at least give the reader some idea of the scatter.*

**Proposed correction:** We will include the standard deviation of the $R^2$ of the regression and of the Pearson correlation coefficient r in the abstract.

2) *Page 4, lines 25-28: The definition of a large lake (ocean-like conditions) is a little vague. Might be useful to have a quantitative definition of what a large lake and a small lake are? Maybe something related to the minimum width of the widest part of the lake?*

It would have been better to provide a quantitative definition of a large lake and a small lake here, to be clear about which lake sizes are expected to give a certain altimetry

accuracy. If the altimetry accuracy was directly related to lake size, we could have been more explicit about the 'large and small' terms in this paragraph. This is however not the case. Large lakes are more likely to give a higher accuracy than smaller lakes, but we found no direct clear relationship between lake size and accuracy. Altimetry accuracy is dependent upon many other factors, like surrounding topography, surface waves, the shape of the water body, the sensor, and the position of altimeter track crossings. However, we mention the minimum lake size of a few hundred meters that can still yield accurate altimetry measurements, only in case all above mentioned conditions are ideal.

**Proposed correction:** Although we cannot be much more explicit in the term 'large and small' lakes, we totally revised section 2.1 to expand our explanation on the different causes of altimetry uncertainty and the complications for small water bodies.

**Revised text for the second paragraph of section 2.1 (starting on line 5, page 4):** 'The estimation of water level time series for small lakes, reservoirs or rivers is very challenging. Due to coarse mission-dependent ground tracks with a cross-track spacing of a few hundred kilometres, larger lakes and reservoirs have a much higher probability to be crossed by a satellite track than smaller ones. Moreover, small water bodies tend to have a relatively big altimeter footprint compared to their size, which will affect the resulting shape of the returning waveform. The diameter of the footprint is mainly influenced by the water roughness (i.e. surface waves) and surrounding topography. In reality, the diameter of the footprint can therefore vary between 2 km over the ocean and up to 16 km for small lakes with considerable surrounding terrain topography (Fu and Cazenave, 2001). These land influences and surface waves within the altimeter footprint can affect the altimeter waveforms and require an additional retracking to achieve more accurate ranges. In order to achieve accurate results for small water bodies, the conditions have to be ideal , meaning a low surrounding topography, low surface waves and perpendicular crossings of the altimeter track and the water bodies shore. In these ideal cases, satellite altimetry has the capability to observe rivers with

a width of about 100-200 m or lakes with a diameter of a few hundred meters. The off-nadir effect is another problem which can occur when investigating smaller water bodies. In general, satellite altimetry measures in the nadir direction, but if the investigated water body is not located in the center of the footprint, then the radar pulses are not reflected in the nadir direction which leads to longer corrupted ranges that must be taken into account (Boergens et al., 2016)'.

**Revised text for fourth paragraph of section 2.1 (starting on line 25, page 4):** 'The quality of the water level time series from satellite altimetry in DAHITI has been validated with in-situ data. For large lakes with ocean-like conditions (such as the Great Lakes), accurate measurements can potentially be achieved with a root-mean-square error (RMSE) as low as 4-5 cm, while for smaller lakes and rivers the RMSE could increase towards several decimeters (Schwatke et al., 2015a). However, no clear relationship was observed between lake size and altimetry accuracy, as the quality of water level time series is not only dependent on the target size, but also on many other factors (e.g. surrounding topography, surface waves, winter ice coverage, the position of altimeter track crossings).'

3) *Page 10, Figure 4: there are large departures in the plot for Lake Nasser – what could cause these? How significant are they?*

These outliers may be caused by time lags between altimetry measurements and Landsat observations in the GSW dataset, as explained in the second paragraph of section 5.3. We cannot correct for this uncertainty, as the GSW dataset did not save the exact dates of the Landsat observations, but only provides the month of observation. Therefore, the time lag between the measurements can be up to one month. In this extreme case of the outliers for Lake Nasser, both water levels were measured in the beginning of the month (2th and 6th day) and were the only measurements available during that month. For the next month, we observed a considerable change in water level. The area and water level observations thus likely refered to different lake conditions, which could be a reasonable cause for the outliers seen in Figure 4d in the manuscript. However, these specific outliers do not have a large influence on the regression as they represent only 5 % of the residuals (n=41). Considerable outliers caused by this effect are rare, as the water level change within a month has to be big, the number of altimetry measurements limited and the difference in timing of the A and h observations considerable. However, we expect this effect to be an important overall contributor to the residuals. To account for the area-level regression uncertainties, we extend the analysis by estimating residual-based confidence intervals (see comment 7).

**We will add the following sentences to the revised version of the paper (page 19, line 8):** 'The outliers in the regression of Lake Nasser (Figure 4d) are expected to be largely induced by this uncertainty. For both outliers, the altimeter measurements were taken in the beginning of the month (2th and 6th day), and the water level changed considerably towards the next month. The Landsat observation therefore likely measured different lake conditions than the satellite altimeter'.

4) *Page 11, lines 14: It would be good to give some information on how the uncertainty was obtained, and how the no data pixels were treated in estimating the points shown in Figure 6. Are the red points likely to be lower bounds on the lake volume? From the Figure, these seems to be the case. If they are lower bounds, then the red shaded area seems to span between this low bound and an estimated upper bound. How are the individual pixels within the MWE converted to an area? Is this simply adding up the number of pixels with a detection of a water surface? Appears to be so based on what I can see, in which case this is a very simplified approach, and better estimates of the upper and lower bounds could be made by considering the part of wet pixels at other times (I note that some discussion on this appears in page 18, reinforcing my interpretation that the simplified approach has been used).*

The uncertainty described here is the uncertainty directly induced by the no data pixels within the maximum water extent (MWE). As described on page 7, line 25, the no data fraction (no data pixels within the MWE / all MWE pixels) was limited to only 5 %. These no data pixels are located within the MWE, and have thus been classified

as surface water at least once over 1984-2015. This means there is a probability that they are water for that specific month. Therefore, the area of these no data pixels is simply added to the observed monthly lake areas, and the upper limit of the volume estimate is subsequently calculated using this upper area limit. To conclude, the red and blue lines in the volume variation plots (figure 6 in the manuscript) are our best volume estimates calculated with observed water area values. If any of the no data pixels within the MWE were covered by water, the estimated volume variation will be somewhere in the red shaded area.

This is indeed a simplified approach, but at least gives an indication of the amount of 'no data' and how this can affect the volume estimations. Techniques that can improve on this limitation are outlined in lines 12-18, page 18, and they got potential for further research. However, for this research we chose to use only direct observations of surface water, as these techniques to reduce 'no data' induce additional uncertainties and their complexity requires a whole new study.

However, we agree that we can explain this uncertainty in more detail. **Therefore we propose to substitute the sentence in line 13-16, page 11 with:** 'The red line displays the best estimate of the volume variation as calculated with observed water classifications in the GSW dataset (i.e. total area of surface water). The red shaded area displays the upper volume boundary on the $V_{GSW}$ estimates, as derived from the GSW dataset pixels classified as no data within the MWE (max 5 %, see section 3.2). These no data pixels could theoretically be covered with water for that month, and this would increase the estimated area. In this case the volume variation estimation would be somewhere within the red shaded area. The upper limit of the red shaded area would thus be reached if all no data pixels within the MWE contain surface water during that particular month.'

5) *Page 14, lines 8-13: Information on data sources seems to be incomplete. Sources for US, Spain and Sudan are given, what about the source for the 2 lakes in Australia?*
You are right, we forgot to mention the source of the 2 lakes in Australia. We will add the link in the revised manuscript. **We will add the following sentence to the revised version of the paper (page 14, line 13):** 'Validation data for Lake Argyle and Lake Eucumbene were obtained from WaterNSW in Australia via http://realtimedata.water.nsw.gov.au/water.stm.'

6) *Page 14, lines 13-14: Seems strange to make the statement that the NRMSE is relatively low taking into account all sources of uncertainty, but there is no discussion about what the sources of uncertainty are that have been considered, or the magnitude of the overall uncertainty. Does this statement mean that the NRMSE is a lot smaller than would be expected given the estimated uncertainty? If so, it suggests a possible error in the uncertainty quantification (e.g. ignoring the impact of serial correlation between the different component uncertainties).*

We fully agree with you, so this sentence should be deleted. This statement is uninformative, as we do not quantify the total uncertainty as induced by all different uncertainties mentioned in the discussion.

7) *Page 15, line 1: Yes, extrapolating beyond the limits of the data will result in higher errors. This is why the uncertainty in the regressed coefficients should be reported. Even then, the uncertainty estimated from the regressed quantities will be a lower bound on the uncertainty in the extrapolation as the estimate is based on the assumption that the fitted function still holds. Possible explanation for the over-estimation of the extrapolated storage for Lake Mead shown in Figure 8 (regressed coefficients are time dependent, or relationship is not as linear as was originally thought), or is the red line shown there within the uncertainty bounds for the original regression?*

We agree that including the regression-based uncertainty may provide useful information for understanding estimation errors. Therefore, we estimated the regression uncertainty based on the standard deviation of the residuals, assuming they have a zero-mean Gaussian distribution $\epsilon \sim N(0, \sigma_\epsilon^2)$. From this residual-based uncertainty,

we derived confidence intervals (CI) around the volume variation estimates.

**Proposed correction:** In the revised manuscript we will substitute the residual term $\epsilon_i$ with the $\frac{1+\alpha}{2}$ and $\frac{1-\alpha}{2}$ quantiles of the Gaussian distribution of $\epsilon$ to estimate a 95 % CI around the regression (see Figure 1).

We revised the old volume plots without a CI (Figure 6 in the manuscript) to volume plots with the 95 % residual-based CI (Figure 2) and the old validation plots (Figure 9 and 10 in the manuscript) to validation plots including the CI (Figure 3 and 4).

This CI represents the uncertainty directly from the standard deviation of the residuals. This is a lower limit of the actual uncertainty, as it does not account for model uncertainty (i.e. it assumes that the fitted linear function holds). However, still an average of 60 - 65 % of the validation volume variations fall inside the 95 % CI for the 18 validation lakes. This suggests that the majority of the uncertainty is captured.

Equation (2) and (3) now estimate the expected value of the volume $E[V_i]$.

**After these equations, on line 24, page 7, we will add the following:**

'Subsequently, a confidence interval (CI) was calculated around the expected value of the volumes calculated with A. The residual term in Eq. (1) was included in the volume calculation (Eq. 3) to estimate the residual uncertainty on the expected volumes calculated with A values. It is assumed that the residuals have a zero-mean Gaussian distribution $\epsilon \sim N(0, \sigma_\epsilon^2)$, where $\sigma_\epsilon$ is the standard deviation of $\epsilon$. To obtain the $\alpha$-probability CI around the expected volume $E[V_i]$, the residual term is replaced by its $\frac{1+\alpha}{2}$ and $\frac{1-\alpha}{2}$ quantiles:

$$CI_{E[V_i]} = \frac{a \cdot A_i^2}{2} \pm \frac{A_i}{2} \cdot \sigma_\epsilon \cdot \phi^{-1}(\tfrac{1+\alpha}{2})$$

Where $\phi^{-1}(\frac{1+\alpha}{2})$ is the inverse of the cumulative density function of the standardized Gaussian distribution (mean = 0, standard deviation = 1) at probability level $\frac{1+\alpha}{2}$. In this research, a 95 % CI has been used.'

It should be noted that the CI for the altimetry volume estimates is not shown in the plots, to keep them readable. They could be simply derived using the same methodology (i.e. by expressing the volume in terms of water level instead of area).

**Additional remark for Lake Mead:** For the extrapolated part of the volume variations we are not sure about the bathymetry of the lake, as we do not know if the estimated bathymetry in the regression will hold for extreme h or A values that are outside the h-A domain of the regression. For Lake Mead, this uncertainty likely caused the overestimation of the volume variation since 1984. This could mean that for the extrapolated part of the regression, the change in water level is in reality less sensitive to a change in lake area than what would have been expected given the found hypsometry (i.e. the slope of the regression is in reality less steep). If the water levels for the whole range of A values were included, the regression would therefore most likely either be explained by (1) a linear regression with a more gentle slope or (2) a regression with decreasing slope for higher A values. However, by comparing the in situ volume variations with the satellite estimations (Figure 8 of the manuscript), we hypothesize that a linear regression would still hold, but with a more gentle slope. This would probably avoid the overestimation as is observed for the extrapolated volumes now.

8) *Page 19, lines 2-5: A non-linear hypsometry relationship shouldn't mean the lake volumes are unreliable. Just that more care and some more maths is needed to derive the volumes. The main issue would be the choice of fitted function, and how this behaves under extrapolation. Given the result shown in Figure 12, a hyperbolic function that becomes roughly constant as area decreases, and linear as area increases would likely be a much better function to fit than a quadratic or a cubic.*

I think there is a misunderstanding about this sentence as we did not clearly formulate it. We do not suggest that volumes of these non-linear lakes are unreliable, but that for these lakes the volumes estimates assuming linear area-level relations are unreliable. So with our current methodology, the calculated volumes are assumed to be unreliable for these lakes.

[Figure]

**Proposed correction, page 19, line 2-3:** 'Only three out of 135 lakes (Tawakoni, Ur-mia and Eagle) showed a clear non-linear area-level relation. For these lakes, volume variations were not estimated'

**Minor comments**

1. *Page 5, line 26 (and elsewhere): might be better to have all acronyms in capital letters.*

Yes thank you, we will check all acronyms and revise this.

2. *Page 9, line 11: "Lakes Powell, Kariba, Mead and Nasser"*

This will be corrected in our revised document.

3. *Page 12, line 6 (and elsewhere): $km^3$ is not a standard SI unit. The equivalent SI unit would be TL (teralitres). Is km3 acceptable?*

Yes this is a HESS guideline, as indicated in the 'Manuscript preparation guidelines for authors'.

4. *Page 12, line 23: "during which time it lost approximately 30 km3"?*

Yes, this is a better formulation. We will revise this in our manuscript.

**References**

Boergens, E., Dettmering, D., Schwatke, C. and Seitz, F.: Treating the hooking effect in satellite altimetry data: A case study along the mekong river and its tributaries, Remote Sens., 8(2), doi:10.3390/rs8020091, 2016.

Fu, L. L. and Cazenave, A.: Satellite altimetry and earth sciences: a handbook of techniques and applications, Academic Press., 2001.

Schwatke, C., Dettmering, D., Bosch, W. and Seitz, F.: DAHITI - An innovative approach for estimating water level time series over inland waters using multi-mission satellite altimetry, Hydrol. Earth Syst. Sci., 19(10), 4345–4364, doi:10.5194/hess-19-4345-

[Figure]

2015, 2015.

[Figure]

[Figure]

Fig. 1. Area-level regressions for Lake Powell (a), Kariba (b), Mead (c) and Nasser (d), with R2 values of respectively 0.99, 0.96, 0.98 and 0.92.

**Fig. 2.** Lake volume variations for Lake Powell (a), Kariba (b), Mead (c) and Nasser (d) using VAltimetry (blue) and VGSW (red).

[Figure]

**Fig. 3.** Validation time series plotted with estimated reservoir volumes for Lake Mead. The black triangle line represents the validation storage as measured using the full lake bathymetry.

[Figure]

**Fig. 4.** Validation time series plotted with estimated reservoir volumes for Lake Powell. The black triangle line represents the validation storage as measured using the full lake bathymetry.

---

## Author Comment (AC2) · 23 Jul 2018

Thank you for your time and interesting comments about our paper.

*It is an interesting paper that promotes the use of remote sensing data. It is also well written and well organised. The authors claim that their study can be useful in the assessment of changes in water availability due to climate change. However, I cannot see how the latter can be achieved based on the material presented. The authors are asked to expand on that thought. In addition, I am not sure how this publication might contribute to the enhancement of knowledge on the physical processes involved. In other words, a link is missing between the paper's main contribution – a new lake and reservoir volume dataset - and the possible applications.*

[Figure]

This paper's main aim is to develop an automatic methodology to calculate volume variations with remote sensing, that improves on current monitoring techniques. However, we realized that a more detailed and explicit explanation on the possible applications of the dataset will indeed improve the paper.

**Proposed correction:** Therefore, we added a much more detailed explanation of the possible applications that will replace the former 'explanation' on line 5-10, page 17:

'The lake and reservoir volume dataset developed here will help to better understand the behaviour and operations of lakes and reservoirs. As the number of reservoirs is still increasing because of growing energy demands, it is crucial to include their effects in (continental and global scale) hydrological models. Zajac et al. (2017) found that the exclusion of lakes and reservoirs often leads to inaccurate downstream discharge estimates. Furthermore, lake or reservoir storage change combined with modelled or observed inflow allows for a better estimation of the outflow (e.g. Muala et al., 2014). These outflow estimates can be used to calibrate hydrological models or estimate hydropower production in areas where in situ observations are lacking. However, due to a lack of storage observations and their availability – often because of commercial reasons -, the parametrization and the representation of lakes and reservoirs in many hydrological models – if at all present - is still highly simplified. Our global lake and reservoir volume dataset over 32 years will be very beneficial to calibrate and validate their parameterisation to mimic their operational behaviour. This will improve our current understanding of lakes and reservoirs, improve their simulations and consequently the simulations in the rest of the river basin. In addition, a better understanding of reservoirs will also likely improve water and energy production projections of these reservoirs under climate change, or under different management scenarios (e.g. changing downstream water requirements, flow legislation, changing inflow due to other activities upstream). Moreover, the area time series developed in this study can be included in models to improve on (often fixed) current area estimates and can furthermore improve estimates of open water evaporation.'

*The authors mention that some lakes showed poor regression between h and A, with one of the reasons being lake size. What are the limits of detection of h(A) relationship from the satellite data (what is the minimum lake area detectable)?*

One of the reasons for poor regressions between h and A is indeed lake size. However, as also explained in AC1 and in the discussion section, many other factors induce uncertainty in the regression (e.g. topography around the lakes, lake shape, position of the altimeter track, wave height, ice coverage, time difference between level and area observation, amount of no data in the GSW dataset). Therefore, we did not found a clear relationship between lake size and the $R^2$ of the regression. A hard limit on the minimum lake size for which the regressions are accurate is hard to give, as it thus depends on many other factors. However, we observed two lakes with an area $< 10km^2$ (Barragem do Caia and Encoro de Salas) that still showed accurate regressions. For these lakes the surrounding conditions were ideal and above mentioned uncertainties had a limited influence.

**Proposed correction: We will add the following sentences in line 18, page 17:** 'However, many other factors than the size of the water body determine the accuracy of the measurement; also surrounding topography, surface waves, winter ice coverage, the shape of the water body and the position of the altimeter track determine the measurement error. This could explain why no clear relationship between lake size and regression accuracy ($R^2$) was observed. Although most lakes with an area $< 10\ km^2$ showed poor regression results, some of these small lakes still returned an accurate regression (e.g. Barragem do Caia and Encoro de Salas). '

*From the satellite altimetry data description we learn that the accuracy of water level time series varies with the lake size, from 4-5 cm for large lakes up to over a meter (several decimetres) for small lakes and rivers. Can this variable error variance be included in the calculation of volume variations?*

The satellite altimetry data description indeed states that the accuracy for large lakes

can potentially be higher than for small lakes. However, as described above and in comment 2 of AC1, the accuracy depends on many other factors. Therefore, we cannot find a relationship between lake size and altimeter accuracy that is clear enough to include the altimetry error in the volume variation estimation.

**References**

Muala, E., Mohamed, Y. A., Duan, Z. and van der Zaag, P.: Estimation of reservoir discharges from Lake Nasser and Roseires Reservoir in the Nile Basin using satellite altimetry and imagery data, Remote Sens., 6(8), 7522–7545, doi:10.3390/rs6087522, 2014.

Zajac, Z., Revilla-Romero, B., Salamon, P., Burek, P., Hirpa, F. and Beck, H.: The impact of lake and reservoir parameterization on global streamflow simulation, J. Hydrol., 548, 552–568, doi:10.1016/j.jhydrol.2017.03.022, 2017.

---

## Author Comment (AC3) · 23 Jul 2018

*After reviewing the paper of lake volume analysis using satellite altimetry, it is found the topic is good and is of great practical interest, because the determination the relationship between volume and water level is very important for reservoir operation and for rational water allocation. The methods employed in the study are also good compared with the traditional methods by surveying. It will become popular for the estimation of lake volume in the remote region, such as mountain rgion. Therefore, it is recommended to publish in the journal.*

Thank you for your interest in our paper and positive comments.

21, 2018.

---

## Author Response (AR2)

**"A global lake and reservoir volume analysis using a surface water dataset and satellite altimetry" by Busker et al.**

**Cover letter to the editor**

**27 November 2018**

Dear Prof. Anas Ghadouani,

I am pleased to submit the second revision report of our paper "A global lake and reservoir volume analysis using a surface water dataset and satellite altimetry" (hess-2018-21).

We really appreciated the additional referee comments. A point-wise overview of these comments and suggestions with our responses and proposed corrections is attached below. Subsequently, a marked-up manuscript version is provided. Please note that revised figures appear on top of their older version.

We hope that this version is now acceptable for publication in the journal and that it may contribute to monitoring techniques of lakes and reservoirs from space.

With kind regards,

Tim Busker

**Response to Report #1 by Anonymous Referee #1**

*The paper has been improved, thought there are still areas where further improvement is in my view needed (see comments below). I think the paper needs some further revision before being ready for publication.*
Thank you a lot for your clear and constructive feedback on our paper. We agree your comments will indeed improve the overall quality of the paper, so they are highly appreciated. Below we first point-wise replied to all your comments, after which we provided track-changes to clearly visualise our proposed corrections to the paper with respect to the most recently submitted manuscript.

**Specific comments**

*1) Page 1, line 19 and elsewhere: While a hard space is included between a number and the unit, the exception is where the unit starts with a superscript. Percentages fall into this category, and the space between the number and the % symbol should be removed.*
Thank you for this comment. We deleted the hard space between the number and the % symbol everywhere in the document.

5 *2) Figure 3: Why use Lake Mead in this figure, seeing as the same results are shown in Figure 4? Could a different lake (not shown in Figure 4) be used? Note that the results for Lake Mead suggest the relationship is slightly non-linear, which means that extrapolating beyond the range shown will result in larger errors, beyond the statistical error shown.*
We agree that the illustration in Figure 3 will indeed become more meaningful if a lake not used in the regression plots (Figure 4) will be used. Therefore, **we propose to substitute Lake Mead (U.S.) for Lake Eucumbene (Australia) in**
10 **Figure 3, without making any changes to the illustration style.**

The results of Lake Mead are indeed slightly non-linear. Since the $R^2$ is 0.98 this effect is very minor for the non-extrapolated volumes. However, this uncertainty indeed becomes more considerable while extrapolating and the confidence and prediction intervals based on regression parameter uncertainty (see comment 3) do not completely account for this
15 model uncertainty. See the response on comment 11 for more discussion on the extrapolation for Lake Mead.

*3) Figure 4: confidence bounds seem to be based on the scatter in the residuals (+- 2 std dev). The confidence bounds should be based on the uncertainty in the regressed coefficients (both intercept and slope), which will result is slightly higher single point prediction confidence bounds near the limits of the plot. The uncertainty in the fit near the centre of the plot will be*
20 *much less than the single point confidence bounds, so this will have much greater variation in the width of the confidence bounds.*
We indeed agree that the confidence intervals (CI) should be based on the regression parameter uncertainty, instead of only the standard deviation of the residuals.

25 **Proposed correction:** For Figure 4, we therefore propose to replace our previously computed residual-based uncertainty bands with the CI around the fitted area-level relationship, based on the uncertainty of the slope and intercept parameters. CI are defined as uncertainty bounds around the expected values of the lake level estimates ($E[\hat{h}_i]$, see the derivation of Eq. (S14) in Sect. S3 of the Supplement):

$$CI_{\hat{h}_i} = \hat{h}_i \pm t_{\alpha/2,n-2} \sqrt{\hat{\sigma}^2 \left[ \frac{1}{n} + \frac{(A_i - \bar{A})^2}{n\,var[A]} \right]}$$

In Figure 4, we also show the prediction intervals (PI), which are defined as the uncertainty bounds around the lake level estimates, thus accounting directly for the residuals scatter (see the derivation of Eq. (S13) in Sect. S3 of the Supplement):

$$PI_{\hat{h}_i} = \hat{h}_i \pm t_{\alpha/2,n-2} \sqrt{\hat{\sigma}^2 \left[ 1 + \frac{1}{n} + \frac{(A_i - \bar{A})^2}{n \, var[A]} \right]}$$

Where $t_{\alpha/2,n-2}$ is the t-score belonging to significance level $\alpha$ for a two-tailed test using n-2 degrees of freedom and $\hat{\sigma}^2$ is the estimated variance of the residuals. It is assumed that the residuals have a zero-mean Gaussian distribution $\varepsilon \sim N(0, \sigma_\varepsilon^2)$, where $\sigma_\varepsilon$ is the standard deviation of $\varepsilon$. In this research, a 95% confidence level has been used (i.e. $\alpha$=0.05).

We computed the PI of estimated volume variations with respect to a reference volume $V_0$ ($\hat{V}_i - \hat{V}_0 = \Delta\hat{V}_i$). These intervals will be displayed in the volume plots (Figure 6) and are based on Eq. (S18) of the Supplement:

$$PI_{\Delta\hat{V}_i} = \Delta\hat{V}_i \pm t_{\alpha/2,n-2} \sqrt{\frac{\left(A_i^2 - A_0^2\right)^2}{4} var[\hat{a}] + \frac{A_i^2 + A_0^2}{4} \hat{\sigma}^2}$$

10   Where $var[\hat{a}]$ is the variance of the estimated slope parameter $\hat{a}$ (as calculated with Eq. (S8) in the Supplement) and $A_0$ is the reference area corresponding to volume $V_0$.

We refer to the Supplement for a derivation of the equations above.

15   The PI around the $V_{GSW}$ volumetric variation estimates have been compared with the in situ volume data used for validation. We found that the 95% PI included the volume changes computed from the validation data 86% of the time, indicating that most uncertainty of the volume estimates is captured by the 95% PI.

**We will substitute the paragraph describing the former residual-based uncertainty bounds (starting on line 7, page**
20   **8), with a paragraph describing the new PI around the estimated volumetric change $\Delta\hat{V}_i$ (on line 29, page 8):** 'A 95% prediction interval (PI) and confidence interval (CI) have been calculated around the linear regression line using Eq. (S13) and (S14) in the Supplement, taking into account the slope and intercept parameter uncertainty and the standard deviation of the residuals. The PI around the linear regression has been propagated to a PI around the estimated volume variations of $V_{GSW}$: $PI_{\Delta\hat{V}_i}$, where $\Delta\hat{V}_i = \hat{V}_i - \hat{V}_0$ and $V_0$ is a reference volume. This PI is based on the variance of the volume difference
25   estimation ($var[\Delta\hat{V}_i]$), as given by Eq. (S18) in the Supplement. The derivation of $PI_{\Delta\hat{V}_i}$ is provided in the Supplement.'

*4) Figure 4d: Uncertainty results for Nasser are dominated by 2 observations. Any idea why these observations should be problematic? What happens if these are removed? Presumably, the confidence bounds will decrease by almost a factor of 2?*
A likely reason for these outliers is the time-lag between the Landsat sensor retrieval, from which we estimate lake area, and
30   the water level observation by the satellite altimeter. For a detailed explanation of this uncertainty, we would like to refer to specific comment 3 in our response to RC1 by Anonymous Referee #1. In the manuscript we submitted in response to this referee comment, we added extra lines for explaining these outliers in Sect. 5.3 (line 12, page 20).

To address this comment, we removed these outliers and indeed found that both the CI and PI decreased by almost a factor 2
35   (47% and 49%, respectively). However, as they represent an uncertainty belonging to the methodology we chose not to exclude them from the regression analysis.

**Proposed correction: we would like to add the following sentences to the paper (line 9, page 10):**
'Two considerable outliers are present in the linear regression of Lake Nasser (Figure 4d). They have a considerable effect
40   on the width of the CI and PI, as these uncertainty bounds decreased by almost a factor two (47% and 49%, respectively) when they were removed. A discussion on the causation of these outliers is given in Sect. 5.3.'

*5) Page 11, line 10: In what way are the linear regressions accurate? How do you define "accurate". One possible definition is that an accurate regression is one that isn't biased. However, by definition, linear regressions are unbiased, so this doesn't help. An R^2>0.8 doesn't mean an "accurate" regression - just means that the residuals are sufficiently small to result in an R^2 of 0.8. Why 0.8? Why not 0.75, or 0.85? Better to just say that the linear regression was used to estimate the volumes in the subsequent analysis*

To avoid future confusion about the term '(in)accurate regression', we deleted it and we will only discuss the regression in terms of the magnitude of the residuals.

We chose the $R^2$ threshold 0.8, as there were only five lakes in the $R^2$ range of 0.75-0.8, and three of them (Tsimlyansk Reservoir, Lake Eagle and Lake Tawakoni) showed clear non-linear area-level relationships. As a result, the volume variations of these non-linear lakes are not well represented by our linear regression-based volume calculation method. On the other hand, all regressions with $R^2 > 0.8$ showed approximately linear area-level relationships and were therefore considered suitable for our methodology. Moreover, three lakes for which we found validation data (Roseires Reservoir, Toledo Bend Reservoir and Lake Almanor) showed a regression $R^2$ between 0.8-0.85. By increasing the threshold to 0.85 we would lose these three opportunities for validation and we would exclude eight other lakes with a clearly linear regression.

**Proposed correction**: **we would like to add the following sentences to the paper (line 12, page 11):**
'The $R^2$ threshold has been set to 0.8 as three out of five lakes in the $R^2$ range of 0.75-0.8 (Tsimlyansk Reservoir, Lake Eagle and Lake Tawakoni) showed clear non-linear area-level relationships, while all lakes with a $R^2 > 0.8$ showed approximately linear area-level relationships. Therefore, the volume variations of lakes with a $R^2 > 0.8$ are well represented by our linear regression-based volume calculation method.'

*6) Page 11, line 14: better to add here that the LvP group will not be used in the following analysis.*
Thank you for this suggestion. Indeed that statement was missing and therefore we added this sentence to the paragraph.

*7) Figure 5: explain logic behind these thresholds? From what I can see given the tables in the appendix, the 0.008 threshold in CV is set based on the minimum CV for the LVG group. The 0.8 threshold in R^2 appears to beaimed at capturing in the LvG group, the point near (0.83,0.82). What is the uncertainty introduced in this classification? Would it be better to have x-axis on a log scale so that distinction between categories can be more easily seen? Also, looks like there is no sharp boundary between Lc and LvG/LvP. With slight change to the threshold, the numbers will change?*
We explained the logic behind the $R^2$ threshold of 0.8 in comment 5. The CV threshold of 0.008 is set, because for lakes with CV < 0.008, considerable level variations generally correspond to very small area changes: the resulting area-level regressions were not robust and satisfactory ($R^2 < 0.6$). As explained on line 29, page 8, this can be explained by the fact that 'the signal-to-noise ratio (SNR) for lakes with a very small CV is likely to be low as errors due to no data, misclassifications and the lake border discretisation with 30 m pixels will mask the actual area variations'. Therefore, we estimate volume variations by assuming the area to be constant. This creates an extra uncertainty as changes in lake area are neglected, but since the CV is very small, this effect will be very minor. If we would increase the 0.008 CV threshold, we would exclude water bodies with good regression results from the regression-based volume calculation. Although we are aware that no sharp boundary can be established between Lc and LvG/LvP, we chose the CV=0.008 threshold as it provided a reasonable separation. We rescaled the x-axis of Figure 5 to a log scale to visualize the threshold in more detail. In addition to the explanation above, **we proposed some textual revisions in paragraph 4.2 to increase its quality.**

*8) Page 11, line 20: mention that these are in group LvG?*
We included the term LvG to make clear these water bodies belong to this category.

*9) Figure 6: The uncertainty bounds shown do not consider the time between values. They seem to be linear connections between the uncertainty in each point. Not an issue for panels a and c, but a significant issue for panels b and d. In the figures, the uncertainty in the gaps is more apparent than the uncertainty in the points. The issue is that the uncertainty in*

*the gaps is at times grossly under-estimated. Either the plot should be modified to help emphasise the uncertainty in the points, or there should at least be some discussion about this in the text.*

We agree that the uncertainty indicators from the 95% prediction interval (grey shaded area) and the no data in the GSW dataset (red shaded area) are point predictions belonging to the $V_{GSW}$ estimates. In order to increase the readability of the plot, these uncertainty estimates have been linearly connected to create the shaded areas shown. We prefer not to add other lines and symbols to Figure 6 to further characterise the uncertainty bands and emphasize the uncertainty in the points, in order to avoid reducing the overall readability of the plot.

Therefore, we will **introduce extra lines in the first paragraph of Sect. 4.3 to empathize that the uncertainty in the gaps has to be interpreted with caution (line 10, page 12): '**Note that both uncertainty indicators (red and grey shaded area in Figure 6) are point estimators of the uncertainty at the $V_{GSW}$ estimates, which are linearly connected to increase the readability of the plot. Therefore, caution should be used when interpreting the uncertainty in the gaps between the observed data points (e.g. between 1987 and 1998 in Figure 6d).'

*10) Figure 7: Might be good to comment on the spatial distribution of the different classes? Seem to be a high percentage of Lc in East USA and Canada for example, with only 2 lakes in the LvG category. West USA seems to be dominated by lakes in the LvG category. Similar in Scandinavian countries. Suggests a significant fraction of the Lc class lie above 45deg N, with from what I can see only 7 LvG lakes. The five southern most lakes (South America) are also either Lc or LvP, reinforcing a possible latitudinal influence? Also a high density of Lc lakes in Africa (south of 15deg N).*

A spatial pattern between the Lc and Lv classes can indeed be seen in North America. This is a very good point and should indeed be noted. All water bodies between 42 and 15º N (in the U.S. and Mexico) were classified as Lv, while 64% of the analysed water bodies on the continent with a latitude above 42º N (primarily in Canada) were classified as Lc. We believe the abundance of Lc lakes above 42º N likely results from a combination of climate characteristics (e.g. low evaporation and precipitation in subarctic climates), human regulations (e.g. the amount of water withdrawals and hydropower generation) and topography. We however do not see this abundance of Lc classifications above 42ºN in Eurasia, were the majority (57%) of the lakes were classified as Lv.

Regarding the distribution of LvP lakes, we indeed observed a slightly higher than average proportion of LvP classifications above 42º N. We think one of the reasons for these low $R^2$ values is that many of these lakes are located in cold climates or at high altitudes and are therefore (seasonally) frozen, which lowers the altimetry accuracy. Moreover, the small size of some lakes can induce considerable land interference in the altimeter footprint, which also lowers the altimetry accuracy (e.g. the four LvP lakes located in Germany and Italy: Thülsfelder Talsperre, 1.7 km$^2$, Hainer See, 4 km$^2$, Lake Resia, 6.6 km$^2$ and Altmuehl See, 4.5 km$^2$) (line 28, page 18). The U.S. indeed has a very high proportion of LvG classifications. This is mainly due to the large stack of Landsat images available from 1984, and their relatively high quality (Sect. 5.2). Therefore, a large range of lake conditions were captured by the regression.

*Concerning the distribution of LvP lakes, how sensitive is this result to the choice of R^2 threshold? If decreased form 0.8 to 0.7 or 0.6 is there a significant change in the spatial distribution? Where are the really bad cases (e.g. R^2<0.3)?*

Lowering the $R^2$ threshold from 0.8 to 0.6 did not considerably change the spatial distribution of LvP lakes, as lakes with a $R^2$ between 0.6 and 0.8 were equally spread over the globe. There were in total 12 really bad cases with a $R^2 < 0.3$, and also they showed an equal spatial distribution.

**Proposed correction**: we would like to add the following sentences to the paper (line 17, page 12):
'A clear spatial pattern between Lc and Lv is observed in North America. An abundance of Lc water bodies were located above 42º N (64%, primarily in Canada), while water bodies between 42 and 15º N (in the U.S. and Mexico) were only classified as Lv. This spatial difference in lake extent variation is potentially caused by a combination of different climate characteristics (precipitation and evaporation), human regulations and topography between Canada and U.S./Mexico. Furthermore, the U.S. has a high proportion of LvG classifications compared to LvP (especially Western U.S.), mainly due to the relatively high number of historical Landsat images from 1984 and their high quality (see Sect. 5.2). Changing the $R^2$

threshold from 0.8 to 0.6 did not considerably change the global spatial distribution of LvP classifications. The spatial distribution of water bodies with extremely low $R^2$ values ($R^2 < 0.3$) was uniform over the whole globe.'

11) Page 16, line 18: Might need to expand the discussion on the behaviour in the extrapolation for Lake Mead? Plot suggests that the magnitude of the variations is about right, but that there is a bias in the result. This potentially means that the pattern of short term variation may look okay, but the amplitude of the variation is likely to be less than the actual variation. In comparison, Lake Powell shows also shows a bias, but this is smaller, and an over-estimation rather than an under-estimation.
We agree that there are inaccuracies in the volume results of both Lake Mead (mostly for 1984-2002, an overestimation) and Lake Powell (1984-1990, an underestimation). However, the reasons for these inaccuracies are different.

For Lake Mead, the $V_{GSW}$ estimates give an overestimation of the volume losses, which is probably mainly caused by the extrapolation of the regression line (between 1984 and 2002) using observed water areas that are larger than those used to fit the regression parameters. We explained the source of this inaccuracy for Lake Mead in RC1, specific comment 7 (in 'additional remark for Lake Mead').

The underestimation of $V_{GSW}$ for Lake Powell between 1984 and 1990 is not caused by an extrapolation. As the regression between h and A returned a very high $R^2$ of 0.99 and covers a very wide range of lake conditions, the inaccuracy is likely to be caused by either the lake area calculations or the USBR in situ validation data. Omission errors in the GSW dataset could have caused this underestimation of $V_{GSW}$, although (Pekel et al., 2016) stated the omission error to be overall less than 5%. The exact causation of this inaccuracy remains debatable.

We agree that we should expand this discussion in the paper, so we **propose to revise the last paragraph of Sect. 5.3 to the following (lines 10-17, page 21):** 'The calculated extrapolated volumes are more uncertain than the non-extrapolated ones, depending on how much the hypsometry relationship changes outside the regression range. In general, if the altimetry measurement period is short compared to the area time series, the range of level and area values captured by the regression line is likely to be small and the volume extrapolation using extreme area observations is expected to be less accurate. This was observed for the Roseires Reservoir and Lake Mead. The validation results for Lake Mead (Figure 8 and 9) indicate that for large areas observed outside the regression range (1984-2002) the found linear hypsometry relationship produces slightly biased volume variation estimates. The extrapolated $V_{GSW}$ volumes slightly overestimate volume losses since 1984 compared to the validation data based on in situ measurements. The slight non-linearity observed in the regression of Lake Mead (see Figure 4c) could be an indication for a stronger non-linear hypsometry relationship in the extrapolated domain. This could be a potential explanation for the slight overestimation of the losses between 1984 and 2002. The additional uncertainty induced by extrapolation cannot be fully addressed by our current uncertainty estimate using the PI. However, including extrapolated volumes hardly changed the overall validation results, because only 15% of the volumes were estimated using extrapolation. Note that the underestimation of $V_{GSW}$ for Lake Powell between 1984 and 1990 is not caused by extrapolation error. This underestimation could be caused by errors in the USBR in situ validation data or by omission errors in the GSW dataset, although (Pekel et al., 2016) stated the omission error to be overall less than 5%. The exact causation of this inaccuracy remains debatable.'

12) Figure 12: Cubic regression is not necessarily the best option. Why not try a hyperbolic form (asymptote for small A near 1271, and for high A increasing linearly)?
We agree with you that using a cubic regression is not necessarily the best option. The polynomial in this case has two critical points where the derivatives of the function are zero (i.e. there is a local maximum and minimum). However, the parameters are so small that the effect of these critical points is negligible (less than 1 cm in water height). In case the lake area would decrease further below 1000 km$^2$ in future, the parameters of the polynomial need to be slightly adjusted, as the volume loss will in this case be slightly overestimated using the current formula. However, we think that for the current domain of observed lake areas, this cubic polynomial fits the data well.

In response to your interesting suggestion, we tried to fit a hyperbolic sine function to the data of Lake Urmia and found that this function form indeed also returned a good fit. Therefore, we **propose to add the hyperbolic sine function to Figure 12.**

We tested the polynomial and hyperbolic functions to provide examples of non-linear area-level relationships; the detailed investigation of non-linear hypsometry relations will be addressed in future research work.

[revised manuscript text omitted]